# Strong linkage between benthic oxygen uptake and bacterial tetraether lipids in deep-sea trench regions

Wenjie Xiao [1,2,3] ✉, Yunping Xu [2] ✉, Donald E. Canfield[1,4],
Frank Wenzhöfer [1,5,6], Chuanlun Zhang[3,7] & Ronnie N. Glud [1,2,4,8] ✉

Oxygen in marine sediments regulates many key biogeochemical processes, playing a crucial role in shaping Earth's climate and benthic ecosystems. In this context, branched glycerol dialkyl glycerol tetraethers (brGDGTs), essential biomarkers in paleoenvironmental research, exhibit an as-yet-unresolved association with sediment oxygen conditions. Here, we investigated brGDGTs in sediments from three deep-sea regions (4045 to 10,100 m water depth) dominated by three respective trench systems and integrated the results with in situ oxygen microprofile data. Our results demonstrate robust correlations between diffusive oxygen uptake (DOU) obtained from microprofiles and brGDGT methylation and isomerization degrees, indicating their primary production within sediments and their strong linkage with microbial diagenetic activity. We establish a quantitative relationship between the Isomerization and Methylation index of Branched Tetraethers (IMBT) and DOU, suggesting its potential validity across deep-sea environments. Increased brGDGT methylation and isomerization likely enhance the fitness of source organisms in deep-sea habitats. Our study positions brGDGTs as a promising tool for quantifying benthic DOU in deep-sea settings, where DOU is a key metric for assessing sedimentary organic carbon degradation and microbial activity.

Marine sediments constitute the largest repository of reactive organic carbon on Earth and serve as crucial archives for reconstructing climate history[1]. In sediments, the oxygen exchange rate is a critical metric for assessing the balance between the preservation and degradation of sedimentary organic matter. This balance directly impacts global atmospheric $CO_2$ levels, thereby influencing long-term climate patterns[2]. Additionally, benthic oxygen availability plays a vital role in regulating numerous biogeochemical processes and profoundly affects the biology and ecology of benthic communities[3,4]. Despite its importance, current methodologies for quantitatively assessing benthic oxygen distribution and availability, especially in deep-sea environments, remain notably constrained[3,5]. This limitation

[1]Department of Biology, HADAL & Nordcee, University of Southern Denmark, 5230 Odense M, Denmark. [2]Shanghai Frontiers Research Center of the Hadal Biosphere, College of Oceanography and Ecological Science, Shanghai Ocean University, 201306 Shanghai, China. [3]Shenzhen Key Laboratory of Marine Archaea Geo-Omics, Department of Ocean Science and Engineering, Southern University of Science and Technology, 518055 Shenzhen, China. [4]Danish Institute for Advanced Study (DIAS), University of Southern Denmark, 5230 Odense M, Denmark. [5]HGF-MPG Group for Deep Sea Ecology & Technology, Alfred Wegener Institute Helmholtz Centre for Polar- and Marine Research, Am Handelshafen 12, 27570 Bremerhaven, Germany. [6]Max Planck Institute for Marine Microbiology, Celsiusstr 1, D-28359 Bremen, Germany. [7]Shanghai Sheshan National Geophysical Observatory, 201602 Shanghai, China. [8]Department of Ocean and Environmental Sciences, Tokyo University of Marine Science and Technology, 26 108-8477 Tokyo, Japan. ✉e-mail: wjxiaocug@126.com; ypxu@shou.edu.cn; rnglud@biology.sdu.dk

underscores the need for innovative indicators like biomarkers, which are preserved in sediments and provide valuable insights into environmental conditions and biogeochemical processes[6].

Among these biomarkers, branched glycerol dialkyl glycerol tetraethers (brGDGTs) are particularly notable for their utility in paleoclimate reconstructions[7–9]. Recent advancements in brGDGT research have unveiled potential connections between these biomarkers and benthic oxygen conditions in sediments[10–12]. Originating from bacteria, brGDGTs consist of two alkyl chains ether-linked to glycerol-3-phosphate (G3P). They exhibit 15 common unique structures, characterized by variations in methylation and cyclization degrees, as well as by their presence in different isomeric forms (e.g., 5- and 6-methyl brGDGTs; Fig. S1)[13–15]. These structural variations are captured by indexes such as the Methylation, Cyclization, and Isomerization indexes of Branched Tetraethers (MBT, CBT, and IR, respectively). Notably, these indexes and their derivatives have demonstrated robust empirical relationships with environmental factors like temperature and pH on both regional and global scales[13,14,16].

One pertinent advancement lies in the expanded knowledge of diverse environmental origins of brGDGTs[17–20]. Initially, brGDGTs were thought to be specific for terrestrial environments, forming the basis for their use as proxies in studying terrestrial paleoclimate[14,21]. However, mounting evidence indicates that brGDGTs also originate from marine environments[18], including estuaries[19], continental shelves[22], the deep ocean[23], and the hadal zone[24]. Marine-derived brGDGTs can come from both planktonic and benthic bacteria[10,25]. While the extent to which these sources contribute to the sedimentary brGDGT pool remains debated, growing evidence supports that marine-derived brGDGTs preserved in sediments, especially in deep-ocean contexts[10,23], are primarily biosynthesized by sediment-dwelling bacteria. These findings present opportunities for utilizing brGDGTs to explore marine benthic environments[10]. However, the environmental factors governing the distribution of marine-derived brGDGTs largely remain to be elucidated.

Another advancement is the growing recognition of the impact of oxygen on the production and distribution of brGDGTs[26]. Oxygen availability plays a vital role in microbial processing and in shaping microbial communities[27], presumably including bacteria responsible for brGDGT biosynthesis[15,28]. Early investigations revealed substantially elevated brGDGT concentrations in the anoxic zones of peats, indicating anaerobic or facultative anaerobic bacteria as the primary producers of brGDGTs[15]. Further studies have examined brGDGTs across oxygen gradients in various ecosystems, including soils[29], peats[30], as well as in both the water column and sediments of lakes[11,12,20] and oceans[25], but provided divergent perceptions. Some studies suggest that oxygen depletion stimulates brGDGT production, while others argue that brGDGT production is favored by oxygen, and that certain brGDGT compounds are produced by aerobic or facultative aerobic bacteria[11,12,30]. Conflicting observations regarding the association of brGDGTs in low-oxygen conditions with either more or less methylated compounds have also been reported[11,20,25]. Recent culture-based studies have not established a clear consensus either[28,31–34]. The intricate relationships between brGDGTs and oxygen observed in both cultural and environmental samples strongly highlight the need for further investigation into the role of oxygen in brGDGT compositions and distributions.

It is reasonable to surmise that benthic oxygen conditions may impact the activity and composition of benthic bacteria, thereby influencing the in-situ production of brGDGTs. However, any specific linkage between benthic oxygen and sedimentary brGDGTs remains unresolved. Quantifying their linkage could be of great importance, as it may reveal the specific oxygen conditions favorable for brGDGT biosynthesis, enhance our understanding of the ecophysiological behavior of marine brGDGT-producing bacteria, and offer a potential molecular approach to resolve benthic oxygen conditions in marine sediments at both modern and geological timescales.

In this study, we investigated brGDGT distributions in dozens of sediment cores collected from the Atacama, Kermadec, and Mariana trench regions, with water depths ranging from 4045 to 10,100 m (Fig. 1A). Notably, each targeted trench region lies beneath water columns with varying levels of productivity, ranging from the oligotrophic Mariana, to the mesotrophic Kermadec, and to the eutrophic Atacama regions[35]. Sediments in these deep-sea trench regions are characterized by extremely high pressure (>40 MPa) and consistent bottom water temperature (ca. 2 °C)[36]. Owing to their relatively long distance from major landmasses, their sedimentary organic carbon exhibits minimal terrestrial influence, predominantly originating from marine sources[24,37]. Additionally, these sites present varied oxygen microprofiles, encompassing a diffusive oxygen uptake (DOU) range of over one order of magnitude (Fig. 1B)[38,39]. These characteristics make our targeted trench systems ideal for investigating the linkage between benthic oxygen conditions and marine-derived brGDGTs.

## Results and discussion
### Distribution and origin of brGDGTs in trench regions
All supporting data are detailed in Supplementary Data 1–4, with the principal data summarized in Table S1. Despite the considerable water depths, our sediment cores exhibit relatively high sedimentation rates attributed to the trench's V-shaped topography, ranging from 0.03–0.08 cm yr$^{-1}$ in the Atacama Trench (AT), 0.03–0.04 cm yr$^{-1}$ in the Kermadec Trench (KT), and 0.04 cm yr$^{-1}$ in the Mariana Trench (MT)[39–41]. Consequently, the sediment cores from these sites, with lengths <40 cm, are estimated to represent a depositional age of <1.2 kyr. The concentrations of brGDGTs at all depths in sediment cores were 215 ± 170 ng g$^{-1}$ (mean ± standard deviation; same hereafter) for the AT, 12 ± 7 ng g$^{-1}$ for the KT, and 15 ± 3 ng g$^{-1}$ for the MT, respectively. When normalized to the total organic carbon (TOC) content, the concentrations were 36 ± 29, 4 ± 1, and 5 ± 1 μg g$^{-1}_{TOC}$ for the AT, KT, and MT, respectively. Notably, the distribution of brGDGTs exhibited relatively small variations across different sediment depths in all three hadal trenches (Fig. 2A), implying that the proportion of individual brGDGT compounds, along with brGDGT-based proxies, remained largely consistent irrespective of whether the values were from the surface, bottom, or averaged across the core.

In all three trench regions, the non-cyclized brGDGT compounds (IIIa, IIIa', IIa, IIa', and Ia) were predominant. They constituted 96 ± 1% in the MT, 78 ± 8% in the KT, and 76 ± 7% in the AT (Fig. 2A; Supplementary Data 2). In contrast, the mono-cyclized (IIb, IIb', and Ib) and di-cyclized brGDGTs (IIc, IIc', and Ic) were much less prevalent, comprising only 2 ± 1% and 1 ± 0% in the MT, 16 ± 6% and 6 ± 2% in the KT, and 17 ± 5% and 7 ± 2% in the AT, respectively. The Principal Coordinate Analysis (PCoA) of Bray Curtis dissimilarities in the brGDGT data revealed compositional and distributional variations at both inter- and intra-trench scales (Fig. 2B). These variations predominantly mirrored the methylation and isomerization degrees of brGDGTs. Specifically, at the inter-trench scale, the MT exhibited significantly higher abundances of 6-methyl brGDGTs (MT, 100%; KT, 73 ± 15%; AT, 56 ± 19%) and hexamethylated brGDGTs (MT, 73 ± 2%; KT, 37 ± 14%; AT, 26 ± 7%), paired with higher IR values (MT, 1.0 ± 0.0%; KT, 0.7 ± 0.1%; AT, 0.6 ± 0.2%), relative to the KT and AT ($p < 0.001$) (Fig. 2A; Supplementary Data 2). At the intra-trench scale, notable differences were observed between non-hadal sites (depth < 6000 m) and hadal sites (depth > 6000 m[42]). The non-hadal sites displayed higher abundances of 6-methyl brGDGTs (KT, 95 ± 4%; AT, 82 ± 5%) and hexamethylated brGDGTs (KT, 59 ± 6%; AT, 36 ± 7%), accompanied by higher IR values (KT, 0.9 ± 0.0%; AT, 0.8 ± 0.1%). Whereas the hadal sites showed significantly lower abundances of 6-methyl brGDGTs (KT, 67 ± 10%; AT, 48 ± 15%) and hexamethylated brGDGTs (KT, 30 ± 7%; AT, 23 ± 3%), along with decreased IR values (KT, 0.7 ± 0.1%; AT, 0.5 ± 0.1%) ($p < 0.001$). The variations in major brGDGT compounds, especially regarding the number of methyl groups and isomeric forms, are

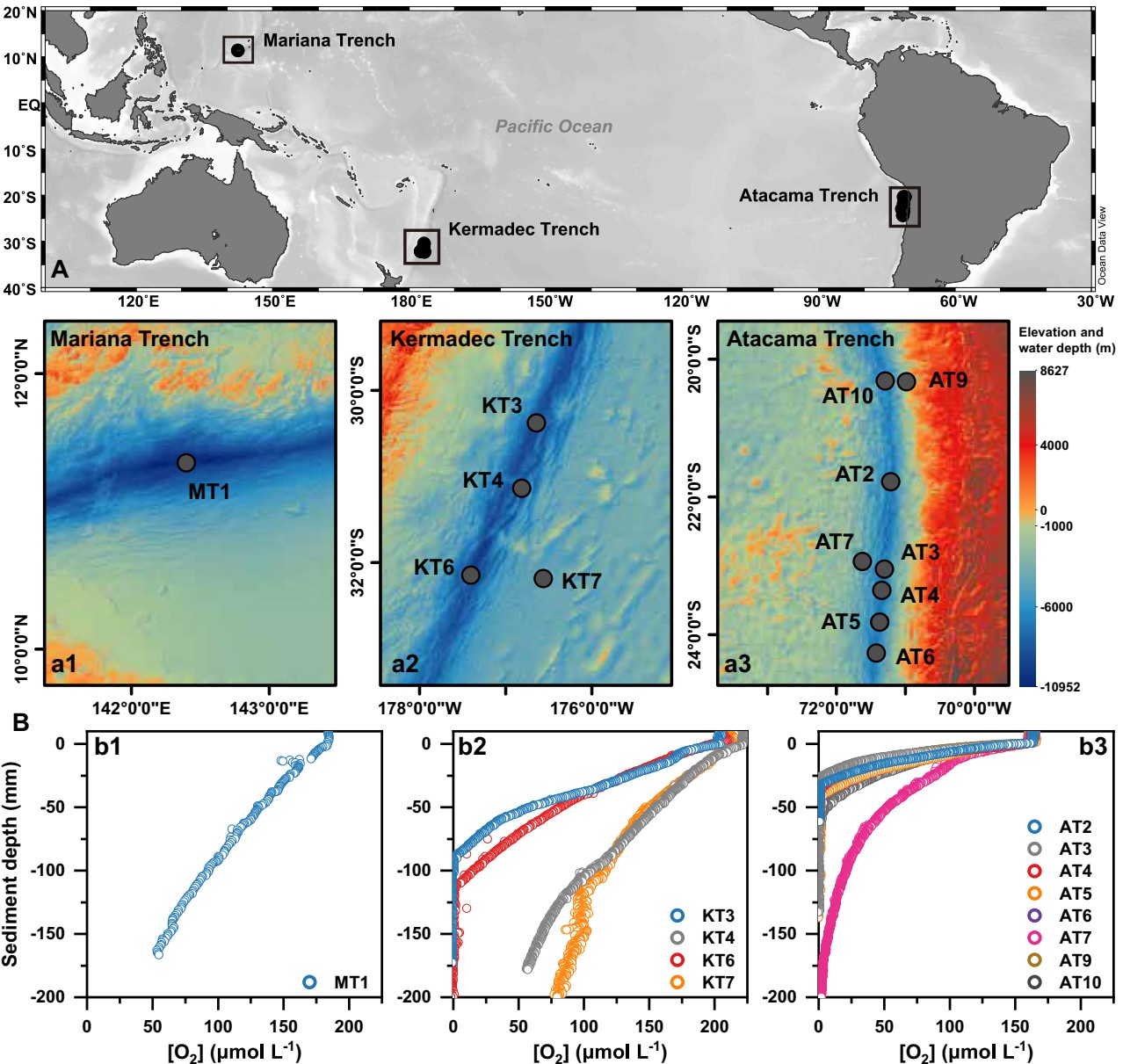

**Fig. 1 | Station sites and benthic oxygen profiles. A** Map of the station sites in the (**a1**) Mariana Trench (MT1), (**a2**) Kermadec Trench (KT3, 4, 6, 7), and (**a3**) Atacama Trench (AT2−7, 9, 10) regions. **B** In situ oxygen microprofiles at the station sites in the (**b1**) Mariana Trench, (**b2**) Kermadec Trench, and (**b3**) Atacama Trench regions. The oxygen data are referenced from Glud et al.[39] and Glud et al.[38].

closely associated with DOU variations (Figs. 2A, 3A), hinting at a tight interrelationship. A comprehensive exploration of this connection will be delineated in the following section.

The branched isoprenoid tetraether (BIT) index[21], a proxy signifying terrestrial influence, generally displays values < 0.15 for open marine sediments and approaches 1 for soils[7,21]. For our trench sites, BIT values ranged from 0.02 to 0.05 (0.03 ± 0.01) in the MT, 0.02 to 0.24 (0.07 ± 0.05) in the KT, and 0.02 to 0.26 (0.07 ± 0.05) in the AT. The ΣIIIa/ΣIIa index, another widely used tool for assessing the origin of brGDGTs[10,18], ranged from 5.58 to 8.35 (7.14 ± 0.98) in the MT, 1.00 to 4.16 (2.00 ± 1.03%) in the KT, and 0.55 to 3.69 (1.45 ± 0.91) in the AT (Fig. S2A; Supplementary Data 2). The ΣIIIa/ΣIIa index generally yields values < 0.59 in soils; while in marine sediments, values are typically 0.59–0.92 for sites with terrestrial influence and typically > 0.92 for sites lacking terrestrial influence[10,24]. Thus, both the BIT and ΣIIIa/ΣIIa indexes imply a predominantly marine origin for the brGDGTs in most trench sediments, although some AT sediments may demonstrate a

slight terrestrial influence. These results are consistent with those obtained from the ternary diagram reflecting the fractional abundance of tetra-, penta-, and hexamethylated brGDGTs (Fig. 3B)[22], as well as the ΣIIIa/ΣIIa vs. IIIa′ diagram (Fig. S2B)[24], both of which support that brGDGTs in our settings are primarily of marine origin. Notably, in-situ production of brGDGTs in marine environments is prevalent[18], a phenomenon we contend that is particular pronounced within our trench regions, distanced from terrestrial sources. This is also supported by the elemental and isotopic signatures of sediment organic carbon[37,43].

In our study, given the close geographic proximity of the different sites within each trench (e.g., KT6 vs. KT7, AT9 vs. AT10, and AT3 vs. AT7; all < 150 km; Fig. 1A), the biogeochemical/physical factors like temperature, salinity, dissolved oxygen concentration, and pH in the water column exhibited minimal differences[44,45]. However, we note striking variations in brGDGT compositions and distributions across sediments (Fig. 2A). Such findings imply that brGDGTs in the trench sites are likely primarily produced within the sediments; and the

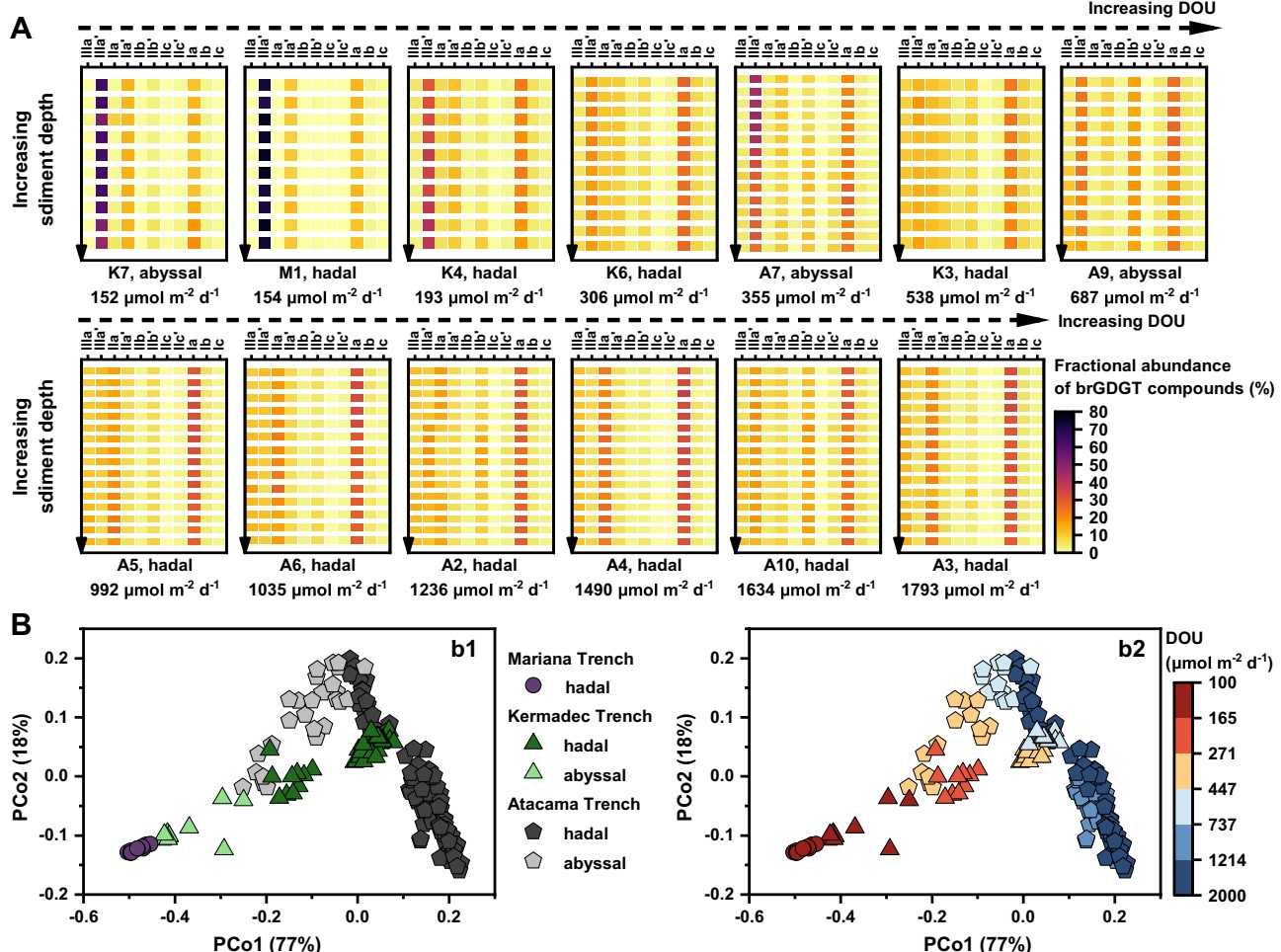

**Fig. 2 | Composition and distribution of branched glycerol dialkyl glycerol tetraethers (brGDGTs) and Principal Coordinates Analysis. A** Heat maps showing the fractional abundance of brGDGT compounds in sediment cores from the Mariana, Kermadec and Atacama trench regions. The colormap visualizes the fractional abundance of brGDGTs in each sample. Diffusive oxygen uptake (DOU) values are labeled for each station site. **B** Principal Coordinates Analysis (PCoA) of Bray Curtis dissimilarities between core sediments from different hadal trench regions based on brGDGT data. **b1** Classification of samples into hadal samples and non-hadal samples from different trench regions, and (**b2**) classification of samples based on DOU values. Source data are provided as a Source Data file.

variability in brGDGTs likely reflect the conditions of benthic environments, such as benthic DOU, a topic that will be discussed in the following section. This inference aligns with a growing body of research suggesting that marine-derived brGDGTs present in sediments predominantly originates from benthic bacteria inhabiting the sediments, particularly in deep-sea settings[10,23].

## Relation between benthic oxygen uptake and brGDGTs

The prevailing concept posits temperature and pH are the key factors influencing the composition and distribution of brGDGTs in terrestrial environments[13,14]. However, in hadal trench sites, where bottom water temperatures are consistently around 2 °C, temperature is not likely the principal determinant affecting brGDGTs. This viewpoint is supported by substantial derivations in reconstructed temperatures using the MBT′$_{5ME}$ index (AT, 4–16 °C, 10 ± 3 °C; KT, 8–19 °C, 13 ± 3 °C; MT, 23 °C)[13]. Despite the absence of in situ sediment porewater pH measurements, pH estimates derived from the CBT$_{5ME}$ index (AT, 5.9–7.2, 6.6 ± 0.3; KT, 6.2–7.2, 6.9 ± 0.3; MT, 6.1–6.1, 6.4 ± 0.2) show an acidic deviation from the typically alkaline pH of hadal bottom seawater and sediment porewater[44,46,47], suggesting that pH might not be the main influence on brGDGTs. To pinpoint the primary determinant of brGDGT variability, we performed correlation analyses comparing both the fractional abundance and concentration of brGDGTs with multiple parameters. These included

surface and bottom water temperatures, geographic variables (latitude, longitude, water depth, and sediment depth), bottom water and sediment porewater oxygen concentrations, diffusive oxygen uptake rate (DOU), concentrations of organic matter (TOC) and total nitrogen (TN), the TOC/TN ratio, and surface net primary productivity (NPP) (Fig. 4 and S3; Supplementary Data 3). Our results unequivocally underscore DOU as the most significant parameter influencing the composition and distribution of brGDGTs across both inter- and intra-trench scales.

DOU denotes the consumption rate of dissolved oxygen within sediment porewater, especially in deep-sea cohesive sediments with minimal fauna activities. This oxygen consumption encompasses aerobic microbial respiration and the reoxidation of reduced products (e.g., $NH_4^+$, $H_2S$, $Fe^{2+}$, $Mn^{2+}$, $FeS$, and $FeS_2$) produced from underlying anaerobic mineralization processes[3,5]. Thus, DOU serves as a crucial metric for assessing the intensity of microbial-mediated diagenetic activity, the degradation rate of sedimentary organic carbon, and the activity levels of benthic communities[3,39]. DOU is influenced by many factors including water depth, TOC, NPP, temperature, bottom water oxygen concentration, faunal activities, and the physical characteristics of the sediment[5]. Our correlation analysis reveals relationships between DOU, NPP, TOC, and brGDGT concentrations. Moreover, beyond DOU, factors like NPP, TOC/TN, and bottom water oxygen concentration, also exhibit correlations with brGDGT fractional

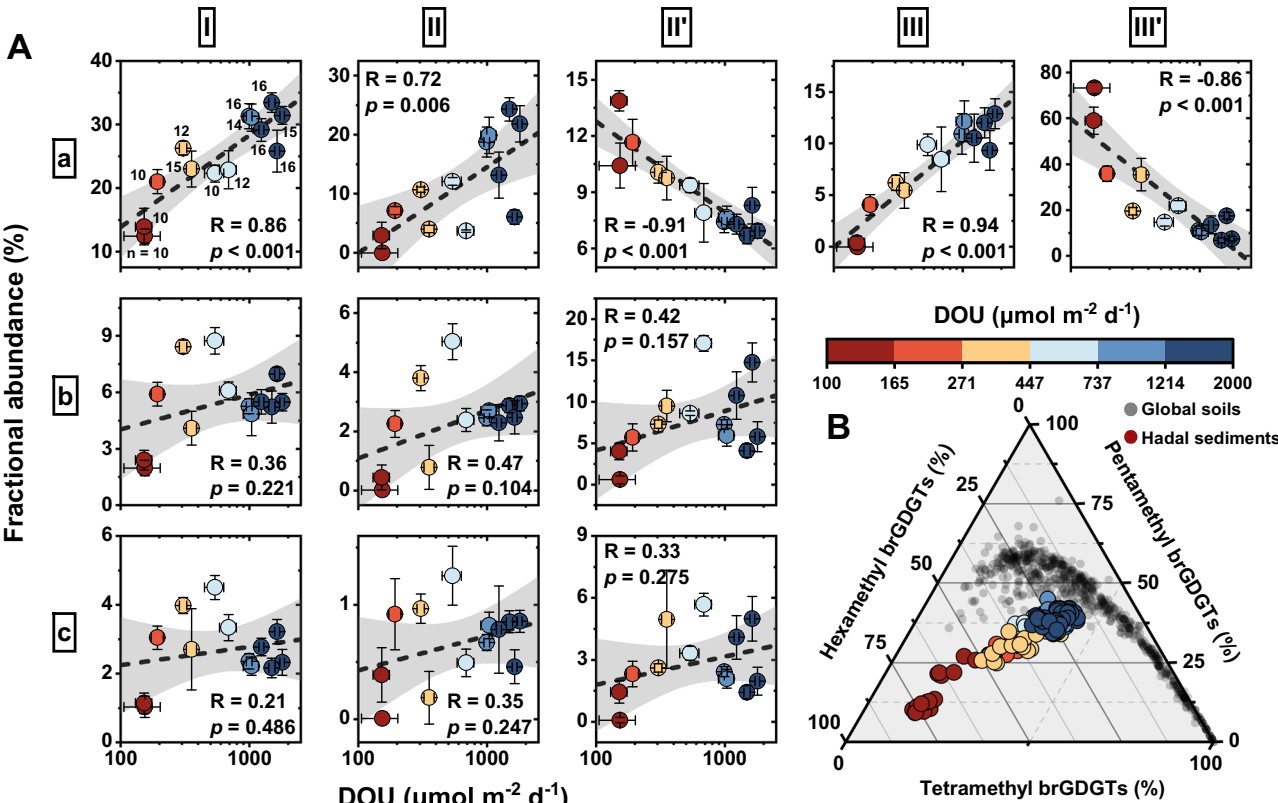

Fig. 3 | **Relationship between branched glycerol dialkyl glycerol tetraethers (brGDGTs) and diffusive oxygen uptake (DOU), and methylation degree of brGDGTs.** **A** Scatterplots comparing the fractional abundance of brGDGTs and DOU. Notice a logarithmic scale of DOU. Linear regression line (black) and 95% confidence intervals (gray band) are shown. Pearson correlation coefficients (R values) are provided for each plot. All reported *p* values result from two-sided Student's t-tests. The circle symbols with error bars denote mean values with standard deviations of variables for each sediment core, with sample sizes specified in the Ia plot. **B** Ternary diagram showing the fractional abundances of tetra-, penta- and hexamethylated brGDGTs in hadal sediments (colored circles) from this study, as well as in global soils (gray circles) referenced from Xiao et al.[24], which integrates data from various sources in the literature. The colormap indicates the DOU values for samples from the trench regions. Source data are provided as a Source Data file.

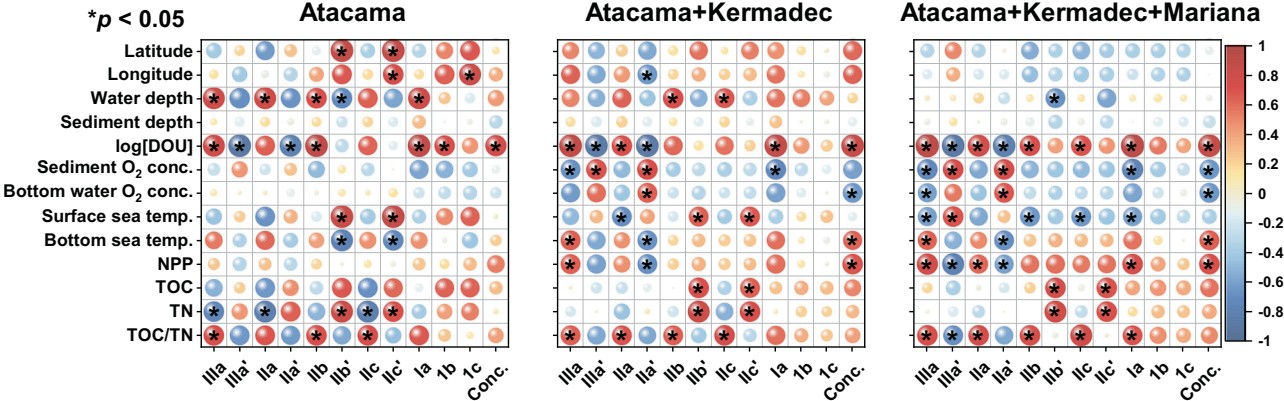

Fig. 4 | **Correlation maps between branched glycerol dialkyl glycerol tetra-ethers (brGDGTs) and environmental variables.** Pearson correlation analysis was utilized to assess the relationships between brGDGTs, both in terms of fractional abundances and concentrations (conc.), and a range of environmental variables. These variables include latitude, longitude, water depth, sediment depth, diffusive oxygen uptake (DOU), dissolved oxygen concentrations in bottom water and sediment, annual sea surface and bottom temperatures, net primary productivity (NPP), total organic carbon (TOC), total nitrogen (TN), and TOC/TN ratio (Supplementary Data 3). Notice a logarithmic scale of DOU. The colormap indicates the correlation coefficients (R values), with red indicating positive correlations and blue indicating negative correlations. The *p* values are derived from two-sided Student's t tests, with *p < 0.05 indicating significance. Due to the limited number of cores from the Kermadec Trench (*n* = 4) and Mariana Trench (*n* = 1), conducting separate correlation analyses for these trenches was impractical. Alternatively, we performed correlation analyses on samples from the Atacama Trench, combined samples from the Atacama and Kermadec trenches, and combined samples from all the trenches. While analyses for surface samples are included here, those for the average values of each core and for the entire core sample dataset are presented in Fig. S3. Source data are provided as a Source Data file.

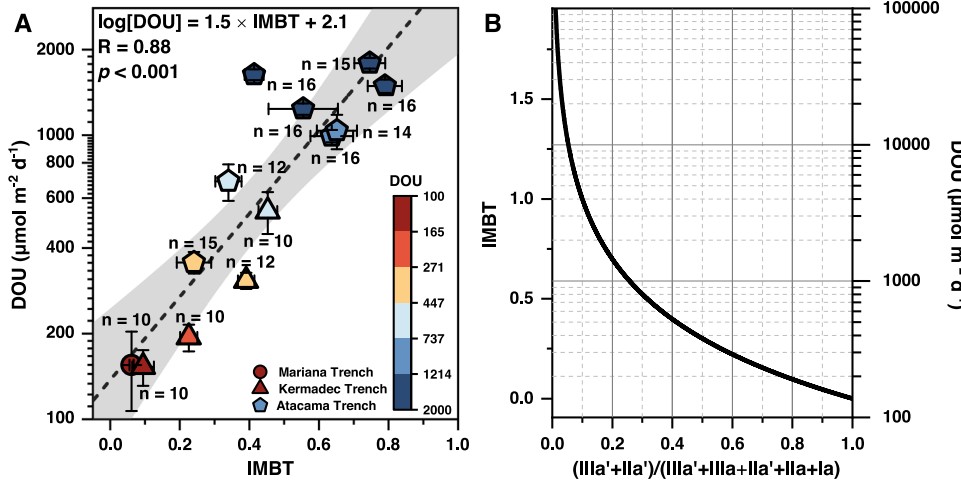

**Fig. 5 | The Methylation index of Branched Tetraethers (IMBT) and sensitivity analysis. A** Scatterplot comparing the IMBT index and diffusive oxygen uptake (DOU). Notice a logarithmic scale of DOU. Linear regression line (black) and 95% confidence intervals (gray band) are shown. Regression equation and Pearson correlation coefficient (R value) are provided. The *p* values result from two-sided Student's *t* tests. The circle, triangle and pentagon symbols with error bars denote mean values with standard deviations of variables for sediment cores from the

Mariana, Kermadec, and Atacama trench regions, respectively, with sample sizes specified within the plot. The colormap indicates the DOU values of the hadal samples. **B** Plot showing how the IMBT index and estimated DOU values change as the proportion of IIIa' and IIa' compounds among acyclic branched glycerol dialkyl glycerol tetraethers (brGDGTs) varies. Source data are provided as a Source Data file.

abundances (Fig. 4 and S3). However, regardless of the inter-trench or intra-trench scale, the relationship between DOU and brGDGTs stands out as markedly more robustness than the other parameters. While previous studies have primarily focused on oxygen concentration and its influence on the abundance and composition of brGDGTs[11,28], our study reveals that with shifts in sediment depth and the accompanying pronounced oxygen concentration decline, the brGDGT compositions see only minor alternations (Fig. 2A). This points to the idea that oxygen concentration may not be the primary determinant affecting brGDGTs in deep-sea sediments. Instead, the role of DOU, which potentially controls the intensity of microbial-driven early diagenesis, emerges as pivotal. These findings may offer insights into the conflicting reports concerning the role of oxygen concentration on brGDGT compositions and distributions as noted in both environmental and cultural studies[11,20,25,28,31–33].

Variability in DOU spans across different sampling sites, showing values of 355–1793 μmol m⁻² d⁻¹ in the AT, 152–538 μmol m⁻² d⁻¹ in the KT, and 154 μmol m⁻² d⁻¹ in the MT[38,39]. Overall, trench axis sites display heightened DOU levels and intensified microbial activities than their adjacent abyssal counterparts[38,39]. Correspondingly, trench axis sites demonstrate markedly lower methylation and isomerization degrees of brGDGTs. In low DOU sites (e.g., MT1 and KT7), brGDGTs are predominantly composed of IIIa' (>50%). Moderate DOU sites (e.g., AT9 and KT3) exhibit a prevalence of IIIa' and Ia, while high DOU sites (e.g., AT3 and AT4) feature Ia and IIa as the primary brGDGTs (Fig. S4). Regression analysis across the three trench regions uncovers a robust positive correlation between DOU and Ia, IIa, and IIIa compounds (r = 0.86, 0.72, and 0.94, respectively; p < 0.001; Fig. 3A). Conversely, IIa' and IIIa' display a strong negative correlation with DOU (r = −0.91 and −0.86, respectively; p < 0.001). These compounds, primarily reflecting the degree of methylation and isomerization, constitute over 70% of total brGDGTs. The cyclic compounds like Ib, Ic, IIb and IIc are much less abundant and exhibit no significant correlation with DOU (Fig. 3A), suggesting that the cyclization degree of brGDGTs is weakly related to DOU. The robust relationship observed between DOU and major brGDGT compounds implies that brGDGTs are primarily produced in situ within sediments and are significantly influenced by the intensity of benthic oxygen consumption or the microbial diagenetic activity.

## A brGDGT-derived index for benthic oxygen uptake

Based on the strong positive correlation between DOU and Ia, IIa, and IIIa compounds, as well as the strong negative correlation between DOU and IIa' and IIIa' compounds, we established the Isomerization and Methylation index of Branched Tetraethers (IMBT) as a quantitative indicator of DOU (Eq. 1).

$$IMBT = -\log[(IIIa' + IIa')/(IIIa + IIIa' + IIa + IIa' + Ia)] \quad (1)$$

$$\log[DOU] = 1.5*IMBT + 2.1 \, (r = 0.88, RMSE = 400 \, \mu mol \, m^{-2} \, d^{-1}, p < 0.001) \quad (2)$$

The IMBT index represents the degree of isomerization and methylation of the acyclic brGDGTs, with lower values indicating a higher abundance of 6-methy penta- and hexamethylated compounds, and vice versa. The IMBT index shows a significant positive correlation with DOU (r = 0.88, RMSE = 400 μmol m⁻² d⁻¹, p < 0.001, Fig. 5A, Eq. 2). Note that both linear and logarithmic forms of the IMBT yield analogous correlation coefficients with DOU within the observed DOU range for the trench sites (100 to 2000 μmol m⁻² d⁻¹). However, we opted for the logarithmic form of IMBT as the linear model would confine the proxy values between 0 and 1. Such a linear model would inherently suggest that the corresponding DOU values are restricted to <4000 μmol m⁻² d⁻¹. Our study, while centered on sediments from deep-sea trench regions, encompasses three distinct trench areas and multiple stations across diverse geographical settings and varied water depths. This breadth reinforces the observed relationship between IMBT and DOU, hinting at its potential universality in deep-sea settings, which are typically characterized by DOU values ranging from 50 to 4000 μmol m⁻² d⁻¹[5]. Collecting additional data from abyssal and bathyal sites would solidify these findings in the future.

We conducted a sensitivity analysis to examine the susceptibility of brGDGT compositions to variations in DOU (Fig. 5B). The results highlight that the brGDGT compositions are especially variable under lower DOU values. For instance, within the DOU range of 100 to 2000 μmol m⁻² d⁻¹, there is a considerable shift in the proportion of IIIa' and IIa' compounds among the acyclic brGDGTs, ranging from 100% to 20% (Fig. 5B). Generally, deep-sea sediments exhibit lower DOU levels

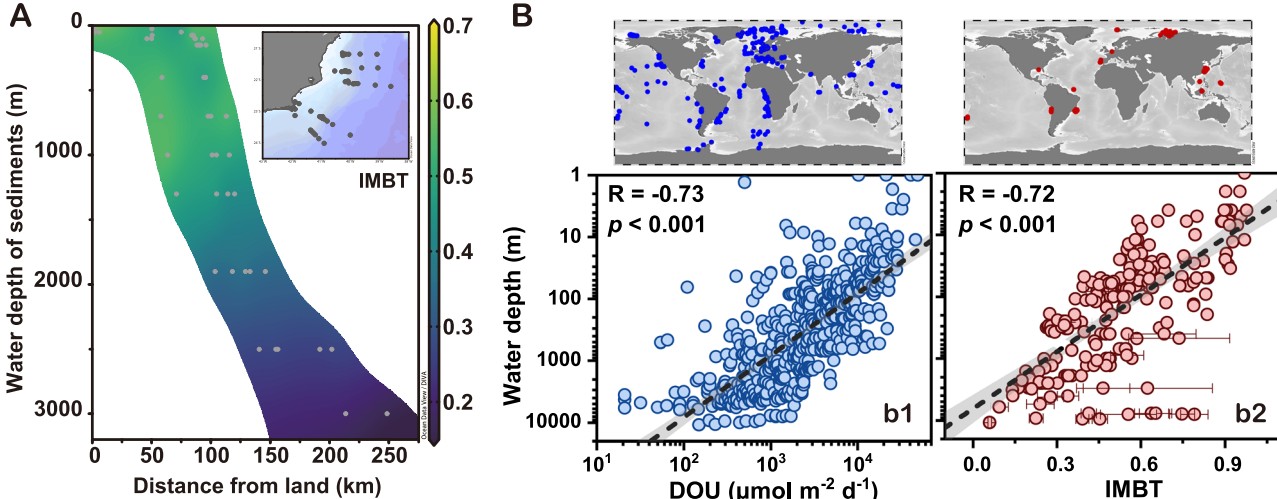

**Fig. 6 | Regional and global evidence of the Methylation index of Branched Tetraethers (IMBT). A** Scatterplot showing how the IMBT index varies with water depth and distance from land in sediments across cross-margin transects in the southeastern Brazilian continental margin. The colormap indicates the diffusive oxygen uptake (DOU) values of the sediments. The data are sourced from Ceccopieri et al.[51]. **B** Scatterplots comparing **b1** DOU and **b2** IMBT with water depth. Notice logarithmic scales of DOU and water depth. Linear regression line (black)

and 95% confidence intervals (gray band) are shown. Pearson correlation coefficients (R values) are provided. The *p* values result from two-sided Student's t-test. The global DOU data are referenced from Jørgensen et al.[5], which integrates information from the literature and databases. The IMBT data are obtained from various studies, as explained in the Data Compilation section under Methods. The circle symbols with error bars denote mean values with standard deviations for sediment cores, with sample sizes provided in Supplementary Data 4.

(50–4000 μmol m$^{-2}$ d$^{-1}$), while coastal sediments tend to have higher DOU levels (2000–100,000 μmol m$^{-2}$ d$^{-1}$)[5]. Therefore, deep-sea sediments would exhibit a broader diversity in their brGDGT distribution patterns, as evidenced by observations from the deep seamount sediments of the Western Pacific Ocean (at depths of 4400 to 5800 m)[48] and the South China Sea (at depths of 700 to 3100 m)[49]. Notably, the variation in brGDGT compositions can become even more pronounced in the deepest hadal trenches, which are characterized by highly heterogeneous environments (Fig. 2A)[42]. It is important to acknowledge that DOU values in our trench region sites spanned from 100 to 2000 μmol m$^{-2}$ d$^{-1}$, and further investigation is warranted to determine the continued validity of the empirical formula between IMBT and DOU in shallow regions with higher DOU values. However, in shallow marine regions, DOU is frequently impacted by benthic fauna activities[3,5], while simultaneously, brGDGTs in these regions tend to be more affected by terrestrial influences and other environmental parameters, like temperature. Consequently, investigating the quantitative correlation between DOU and IMBT in shallow marine settings is likely to be more complicated.

Furthermore, we compared the IMBT index with other commonly used brGDGT indexes, including MBT'$_{5ME}$, CBT$_{5ME}$, DC' (degree of cyclization)[50], IR, and CBT'[13] (Fig. S5 and S6). The analysis revealed that IMBT strongly correlates with indexes related to brGDGT methylation and isomerization (e.g., MBT'$_{5ME}$ and IR; absolute r values, |r| > 0.85), while showing much weaker correlations with cyclization indexes (e.g., CBT$_{5ME}$ and DC'; |r| < 0.35). This pattern extends to DOU, which exhibits strong correlations with methylation and isomerization indexes (|r| > 0.70), but only minimal correlations with cyclization indexes (|r| < 0.30) (Figs. S5, S6). Notably, the correlation between DOU and IMBT stands out as the most pronounced among these indexes. Considering the established association of methylation indexes with temperature and isomerization indexes with pH[13,14], the influence of DOU is crucial in evaluating the reliability of methylation-based temperature and isomerization-based pH indexes, especially in marine settings characterized by predominant in-situ brGDGT production.

### Supporting evidence for the IMBT

To our knowledge, no prior research has simultaneously examined both brGDGTs and DOU. This leaves us dependent on indirect insights

from existing literature to gauge the applicability and validity of the IMBT index. In a regional case study of the southeastern Brazilian continental margin, where brGDGTs in 46 core-top sediments have been reported along cross-margin transects spanning water depths from 25 to 3000 m[51], the region demonstrated minimal terrestrial influence and predominantly in situ marine brGDGT production[51]. As water depth increases and distance from the coastline grows, it would be reasonable to conclude that benthic DOU diminishes (Fig. 6A). Correspondingly, the IMBT index consistently demonstrates a clear declining trend. Furthermore, on a global scale, a systematic decrease in benthic DOU is observed with increasing water depth[5], aligning with the decreasing values of the IMBT index (Fig. 6B).

Thus, the parallel decrease in both the IMBT index and DOU with increasing water depth, observed at both the regional and global scales, hints at a strong linkage between the IMBT index and benthic DOU in marine settings. Notably, data from trench sediments exhibit a deviation from the global correlation between IMBT and water depth (Fig. 6B), owing to the relatively heightened microbial dynamics of hadal trenches which exhibit elevated DOU levels compared to adjacent abyssal plains[38,39]. These observations bolster the credibility of the IMBT index, underscoring its value as an indicator of DOU variations and suggesting its broad applicability across different spatial scales. The global correlation between IMBT and water depth (Fig. 6B) hints at a potential role of hydrostatic pressure on IMBT. However, the elevated IMBT values in trench sediments, which experience significantly higher hydrostatic pressures compared to nearby abyssal sediments (e.g., KT6 vs. KT7, and AT3 vs. AT7; Supplementary Data 2), cast doubt on this notion. The paucity of research on how pressure influences brGDGTs hinders a clear assessment of its impact on IMBT.

### Potential mechanism underlying the IMBT

The empirical relationships between brGDGTs and environmental parameters, whether driven by the direct physiological adaptations of source bacteria to changing environments, and/or the indirect restructuring of bacterial communities, remain an open question[52]. Several biogeochemical and microbial studies have investigated the sites examined in this study[38,53–55]. Our trench core sediments display redox stratification into oxic, nitrogenous, and ferruginous zones[53,54].

Despite bathymetric and depositional heterogeneity, microbial communities exhibit high similarities among sediments sharing similar redox stratifications within each trench, while vary strongly with sediment depth following redox gradients[55]. In contrast, our examination of brGDGT compositions reveals minor variations along vertical redox gradients but distinct differences between different trenches and sites (Fig. 2A). These findings prompt speculation that brGDGT variations might not be primarily driven by bacterial community shifts within trench sediments.

Molecular dynamics simulations suggest that increased methylation in brGDGTs can enhance membrane fluidity[56]. While terrestrial brGDGT-producing bacteria have been shown to adapt to temperature by synthesizing brGDGTs with varied methyl branches[14,31,32], there is limited insight into the ecophysiological behaviors of marine brGDGT-producing bacteria. In trench region sediments, despite consistently cold bottom water temperatures (ca. 2 °C), notable variations in methylation and isomerization degrees of brGDGTs primarily correspond to DOU rather than temperature, suggesting that temperature is not the predominant influencer of IMBT in these settings, as discussed above. In the deep ocean, DOU levels typically reflect microbial diagenetic intensity[3,39]. The correlations observed across our study sites between NPP, TOC, DOU, and brGDGT concentrations (Fig. 4 and S3) suggest a likely connection between a decreased supply of organic carbon and lower concentration of brGDGTs, alongside reduced DOU levels—indicative of diminished microbial diagenetic activity[3,5]. Further, there is evidence indicating that methylation in membrane molecules aids organisms to cope under nutrient or energy constraints[57]. For example, 3-methylhopanoids promote cell viability under nutrient scarcity[58], and increased backbone methylation appears in archaeal GDGT lipids during stationary phases and in nutrient-deprived sediments[59]. Hence, we propose that increased methylation and isomerization of brGDGTs may enhance the fitness of source organisms in deep-sea habitats under nutritional limitations, which usually coincide with reduced microbial activity and lower DOU levels[3,5]. Compared to methylation, isomerization of brGDGTs may be even more influential, as evidenced by the opposing trends observed in IIIa and IIIa' with respect to DOU, while consistent trends are observed in IIIa' and IIa' (Fig. 3A). However, our understanding of how brGDGT isomerization affects membrane properties remains limited. To comprehensively discern how marine brGDGT-producing bacteria modulate their methylation and isomerization in response to microbial diagenetic intensity, further investigations involving environmental experiments, incubation tests, and molecular simulations are essential.

## Implications and significance of the IMBT

The observed correlation between brGDGTs and DOU within our trench region sites implies that marine-derived brGDGTs preserved in deep-sea sediments are primarily produced by benthic bacteria within sediments and are strongly related to benthic oxygen consumption or microbial diagenetic activity. Our results challenge the conventional view that brGDGTs are mainly produced by anaerobic or facultatively anaerobic bacteria[15,60]. Instead, our study indicates that aerobic or facultatively aerobic bacteria in marine sediments can also produce brGDGTs. This insight aligns with recent studies conducted in lakes and pure cultures[11,31,32], expanding our knowledge on brGDGT production across diverse environments. Our findings underscore the pivotal role of DOU in modulating brGDGT methylation and isomerization patterns, which are typically associated with temperature and pH[13,14], respectively. In marine settings where in-situ brGDGT production is predominant, accounting for the effect of DOU becomes essential for evaluating methylation-based temperature and isomerization-based pH indexes[13,61] (Fig. S5 and S6). This research advances our understanding of brGDGTs in marine environments and promotes their application in paleoclimate reconstructions.

Measuring DOU in benthic oceans involves three primary methodologies: sediment incubations, flux calculations derived from oxygen profiles, and the eddy covariance technique[16,49]. These methodologies are commonly constrained by their reliance on specialized and costly in situ analytical instrumentation, which limits the scope and frequency of global ocean measurements. Moreover, they provide only insights into current conditions and are not applicable in studying geological DOU variations. Our study presents an approach using brGDGTs from sediment archives as a deep-sea DOU indicator. The practicality of collecting sediments followed by laboratory lipid analysis is particularly beneficial for broad deep-sea investigations. Furthermore, with brGDGTs traceable back at least millions of years[62], they provide potential for exploring geological DOU variations. In summary, this study pioneers the innovative IMBT index, based on brGDGTs, to quantitatively assess benthic DOU in deep-sea settings, which serve as a key metric for evaluating the degradation and preservation of sedimentary organic matter and for gauging the activity levels of benthic communities.

## Methods
### Study area
The AT is the deepest sector of the Peru-Chile Trench that is formed by the subduction of the Nazca Plate underneath the South American Plate and has a maximum depth over 8000 m[63]. Geographically, it is in close proximity to the Atacama Desert and is characterized by a surface ocean with relatively high net primary productivity (NPP, 300–450 g C m$^{-2}$ yr$^{-1}$). This elevated productivity is primarily attributed to the intense upwelling phenomena occurring in the area. Notably, the Atacama Desert is one of the driest regions globally, leading to the predominantly wind-driven transport of terrestrial materials rather than significant contributions from riverine run-off[64]. The KT is recognized as the fifth deepest trench globally and is formed by the subduction of the Pacific Plate underneath the Indo-Australian Plate[42]. It has a maximum water depth exceeding 10,000 m and extends from approximately 26°S to 36°S near the North Island of New Zealand. The southern end of the trench is about 160 km from New Zealand, and receives low inputs of terrestrial materials[65]. The KT is characterized by oligotrophic to moderately productive waters (NPP, 100–150 g C m$^{-2}$ yr$^{-1}$). The MT is formed as the subduction of the Pacific plate beneath the eastern edge of the Philippine Sea plate[66]. It has a total length of ca. 2500 km and a mean width of 70 km. The Challenger Deep, as the deepest point in the world's oceans, is located on the southern rim of the MT and has a water depth of ca. 11,000 m. The MT is overlain by extremely oligotrophic waters (NPP, <60 g C m$^{-2}$ yr$^{-1}$).

### Sample collection
Waters, sediments, and benthic organisms were collected from the KT, AT, and MT regions during three research cruises aboard the R/V Tangaroa (November to December 2017), R/V Sonne (March 2018), and R/V Zhangjian (December 2016 to February 2017), respectively. A total of 13 sediment cores were selected for GDGT measurements. These cores included four from the KT region (K3, K4, K6 and K7; water depth, 6080–9560 m; core length, 25–40 cm), eight from the AT region (A2–7, A9 and A10; water depth, 4045–8090 m; core length, 10–35 cm), and one from the MT region (M1; water depth, 10,840 m; core length, 11 cm) (Fig. 1A; Supplementary Data 1). For sediment samples from the KT region, the sediment cores were sliced at 1 cm intervals for the top 2 cm, 2 cm intervals for the depth range of 2–10 cm, and 5 cm intervals for depths greater than 10 cm. For sediment samples from the AT region, the sediment cores were sliced at 1 cm intervals for the top 10 cm, 2.5 cm intervals for the depth range of 10–20 cm, and 5 cm intervals for depths greater than 20 cm. For sediment samples from the MT region, the sediment cores were sliced at 1 cm intervals throughout the core. All collected samples were immediately placed in freezers set at −20 °C until further analysis.

The detailed information on the samples has been summarized and documented in the Supplementary Data 1.

## Lipid extraction

Sediment samples were freeze-dried and homogenized for GDGT analysis. About 2–3 g of sediment was subjected to ultrasonic extraction using a 3:1 (v:v) mixture of dichloromethane and methanol, with three extraction cycles lasting 15 minutes each. A known amount of $C_{46}$-GTGT was added as an internal standard prior to extraction. The resulting extracts were centrifuged at 1500 $g$ for 5 min. The supernatants were transferred to pre-combusted glass bottles and dried under a stream of nitrogen. Subsequently, the extracts were separated into non-polar and polar fractions using silica gel columns (60–100 mesh) with n-hexane and dichloromethane:methanol (1:1, v-v), respectively. The polar fraction containing the GDGTs was purified with a 0.45 μm PTFE filter to remove any remaining particles.

## Lipid analysis

GDGTs were analyzed using an Agilent 1260 series High-Performance Liquid Chromatography (HPLC) system coupled with an Agilent 6135B quadrupole Mass Spectrometer (MS) with an Atmospheric Pressure Chemical Ionization (APCI) source. The separation of GDGT compounds followed a modified method proposed by Hopmans et al.[67]. For this purpose, two UPLC columns (BEH HILIC columns, 2.1 × 150 mm, 1.7 μm, Waters) were connected in series. The mobile phase consisted of hexane as solvent A and hexana:isopropanol (9:1, v-v) as solvent B. A 25-min wash using 0% A was initially performed, followed by an elution gradient. The elution gradient involved a transition from 82% A to 65% A over 50 min, followed by a 10-min transition to 100% B, which was maintained for 20 min. Subsequently, a 10-min gradient change of the B component to 18% was implemented and maintained for 30 min to achieve equilibrium. The flow rate was set at 0.2 ml min$^{-1}$, and the column temperature was maintained at 30 °C. In terms of mass spectrometry, the Selected Ion Monitoring (SIM) mode was employed to detect specific ions. The ions monitored included $C_{46}$-GTGT ($m/z$ = 743.6), Ia (1022.0), Ib (1020.0), Ic (1018.0), IIa and IIa' (1036.0), IIb and IIb' (1034.0), IIc and IIc' (1032.0), IIIa and IIIa' (1050.0), IIIb and IIIb' (1048.0), IIIc and IIIc' (1046.0), and crenarchaeol (1292.3). Following data acquisition, manual integration using Agilent MassHunter Qualitative Analysis software (version 10.0) was performed, and compound identification was carried out based on retention time and peak order. Quantification of compounds was accomplished by comparing the peak areas of the internal standard $C_{46}$-GTGT with those of the target compounds. Note that compounds IIIb, IIIb', IIIc, and IIIc' were not reported in this study due to their extremely low abundance or being below the detection limit. The resulting quantification values were reported as dry weight of sediment (ng g$^{-1}$ dws).

## Data compilation

The in situ oxygen microprofile data, which are used to calculate bottom water and sediment dissolved oxygen concentrations, as well as DOU values, are referenced from Glud et al.[39] and Glud et al.[38]. The KT and AT samples analyzed in this study are obtained from the same location as the in situ oxygen measurements. However, the MT samples were collected from a different location (11.43°N, 142.36°E, 10,840 m) that differs from the in situ oxygen measurement site (11.34°N, 142.43°E, 10,850 m). Nevertheless, both sampling sites are geographically close to each other and situated within the Challenger Deep. The data on TOC, TN and the TOC/TN ratio are obtained from Xu et al.[37] and Xiao et al.[24]. Surface and bottom water temperature data are extracted from the WOA18 0.25 degree data[68]. Note that the time periods considered are annual, and the time spans encompass all the data utilized in WOA18, irrespective of the year. The NPP data for the surface waters above the station sites are derived from the standard Vertically Generalized Production Model, using remote sensing data

spanning from 2004 to 2018[69]. To explore the relationship between the IMBT index and water depth, we compiled brGDGT data from various studies, including Cao et al.[70], Cao et al.[71], Ceccopieri et al.[51], Crampton-Flood[72], Crampton-Flood[73], Crampton-Flood et al.[74], De Jonge et al.[19], De Jonge et al.[16], Li et al.[75], Sinninghe Damsté[22], Soelen et al.[76], Wang et al.[49], Warden et al.[77], Zell et al.[78], and the present study. Note that brGDGTs in the surface sediments of the KT and AT have been reported in Xu et al.[79], and brGDGTs in the core sediments of the MT have been reported in Xiao et al.[24]. All the aforementioned data have been documented in Supplementary Data 4.

## Proxy calculation

The BIT index was calculated according to Hopmans et al.[21]. The ΣIIIa/ΣIIa index was calculated following the definition of the IIIa/IIa ratio by Xiao et al.[18], which combined the 5- and 6-methyl isomers to correspond to traditional chromatographic approaches[13,14]. Note that the calculation of ΣIIIa/ΣIIa values excludes the 7-methyl isomers[80]. The IR index was determined based on the approach outlined by De Jonge et al.[19]. The $MBT'_{5ME}$ and $CBT_{5ME}$ were calculated following the approach outlined by De Jonge et al.[13]. The IMBT index was calculated according to this study (see Eq. 1). These calculations involve utilizing the fractional abundance of brGDGTs, represented by roman numerals, and the isoprenoid GDGT crenarchaeol (Cren), which is a specific GDGT associated with Thaumarchaeota[81]. The structures of these GDGTs can be found in Fig. S1. The resulting values of these indexes, along with reconstructed values of environmental parameters (if applicable), have been documented in Supplementary Data 2 and 3.

$$BIT = (Ia + IIa + IIIa + IIa' + IIIa'')/(Ia + IIa + IIIa + IIa' + IIIa' + Cren) \quad (3)$$

$$\Sigma IIIa/\Sigma IIa = (IIIa + IIIa')/(IIa + IIa') \quad (4)$$

$$IR = (IIa' + IIb' + IIc' + IIIa' + IIIb' + IIIc')/(IIa + IIa' + IIb + IIb' + IIc + IIc' + IIIa + IIIa' + IIIb + IIIb' + IIIc + IIIc') \quad (5)$$

$$MBT'_{5ME} = (Ia + Ib + Ic)/(Ia + Ib + Ic + IIa + IIb + IIc + IIIa) \quad (6)$$

$$CBT_{5ME} = -\log[(Ib + IIb)/(Ia + IIa)] \quad (7)$$

## Statistical analysis

The maps in this study were created utilizing Ocean Data View (ODV version 5.6.3) and ArcMap version 10.7 software. Principal Coordinates Analysis (PCoA) was conducted using the vegan package in R version 4.2.1, with the Bray Curtis method employed to generate the distance matrix. To investigate variations in brGDGT compositions and related indexes across different trench sites, one-way analysis of variance (one-way ANOVA) was performed using SPSS version 22.0 software. The significance level for determining statistical significance was set at $p < 0.05$, unless otherwise specified. Pearson correlation analysis was employed to assess the associations between variables, with Pearson correlation coefficients ($r$ values) reported to quantify the magnitude and direction of linear correlations. The molecular structures of brGDGTs were generated using ChemDraw version 19.0 software, while other graphical representations were generated using OriginPro version 9.8.0.200 software.

## Reporting summary

Further information on research design is available in the Nature Portfolio Reporting Summary linked to this article.

# Data availability

All the data supporting the findings from this study are provided in this paper and its Supplementary Information. The WOA18 0.25 degree

dataset is available at https://www.ncei.noaa.gov/products/world-ocean-atlas. The net primary productivity (NPP) dataset, based on the standard Vertically Generalized Production Model, is available at http://orca.science.oregonstate.edu/1080.by.2160.monthly.hdf.vgpm.m.chl.m.sst.php. Source data are provided as a Source Data file. Source data are provided with this paper.

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

## Acknowledgements

This work is financially supported by the National Natural Science Foundation of China (42206040 to W.X., 42276033 to Y.X.), the Danish National Research Foundation grant DNRF145 via the Danish Center for Hadal Research (to R.N.G.), and the Shenzhen Key Laboratory of Marine Archaea Geo-Omics, Southern University of Science and Technology (Grant ZDSYS201802081843490 to C.Z.). W.X. acknowledges support from the HORIZON-MSCA-2023-PF-01-01 Marie Curie Postdoctoral Fellowship (project 101153049, SOCGDGT). Samples were obtained during cruises by the *RV Sonne*, cruise SO261; *RV Tangaroa*, cruise TAN1711; and *RV Zhangjian*, cruise 11,000 m sea trial. We thank the captain(s), crew(s), and scientific personnel for their excellent support in obtaining these samples.

## Author contributions

W.X., R.N.G., and Y.X. designed the study; R.N.G., F.W., and Y.X. organized the cruises and collected samples; W.X. conducted the experiment and data analysis; W.X., R.N.G., D.E.C., Y.X., and C.Z. discussed the results and implications; W.X. and R.N.G. wrote the manuscript with help from coauthors; All authors read, commented, edited, and approved the final version of the manuscript.

## Competing interests

The authors declare no competing interests.
