## [Peer Review File · Nature Communications]

Strong linkage between benthic oxygen uptake and bacterial tetraether lipids in deep-sea trench regionsREVIEWER COMMENTS

Reviewer #1 (Remarks to the Author):

The manuscript titled “Strong linkage between benthic oxygen uptake and bacterial tetraether lipids in deep-sea trench regions” presents data on the distribution of branched glycerol dialkyl glycerol tetraethers (brGDGTs) in sediment cores from the Atacama, Kermadec, and Mariana trench regions. Within these regions, terrestrial input is limited, pressure is high, and bottom waters are consistently similar temperatures, which allows the authors to make general arguments in the manuscript regarding the relationship between oxygen and brGDGTs in marine sediments. The authors identify a strong relationship between diffusive oxygen uptake, which measures the microbial diagenetic intensity.

The manuscript is well written, and the sample and data collection methods follow the typical methodologies used to obtain this type of data. The data presented is novel, and the implications are intriguing as the data may explain why previous studies of brGDGTs distribution in sedimentary environments have yielded conflicting results and inferences about the favorable conditions for bacterial synthesis of brGDGTs and the role oxygen availability plays on the production and distribution of these biomarkers. The paper should be published; however, I suggest a few improvements to the manuscript below.

Suggested Edits:

1. Several figures are overcomplicated, making them difficult to read. I also struggled to see how the data presented applied to what was discussed in the text. For example, Fig. 2A is visually appealing, but it was difficult to read and make interpretations from the heat maps. As Fig. 2A is organized, the instinct is to read the heat maps as though they represent increasing DOU across some horizontal continuum, like along a cross-section. Instead, the two rows are unrelated to each other (the bottom row shows data from the AT collected at different locations in that trench region, whereas, the top row displays the data from one sample from the MT and 4 samples from the KT that were collected from different locations in that trench region). The data are unrelated and not meant to be correlated, nor do they represent change across one cross-section.

It was also difficult to see that the sediment depth arrows represent depth into the core. After looking at the figure for a while, I rotated the paper so that the brGDGT labels were at the top of the maps and the arrow indicating the sediment thickness pointed down as it would when reading data from cores. It might be easier for an audience to read if the heat maps are rearranged to read from top to bottom as you would with data in a core sample. There should also be a way to indicate that these maps are not meant to correlate with each other (as you would when creating a fence diagram or correlating strata).

Unfortunately, I am still unable to identify the relationship you are describing when you reference the figure in line 143. There may not be a better way to convey this information in a figure, which can be challenging. I frequently referred to the data chart in the SI material to gain an understanding of what

was being displayed in the figure.

There is a similar issue with the correlation maps in SI Fig. 3. The figure is responsible for displaying too much information and requires an extended amount of time to decipher what is being displayed. I found the SI table with the environmental data helpful as I looked through this figure.

2. The figures currently in the supplemental information should be included in the main manuscript because they were crucial to understanding the other images in your paper, and they are cited as evidence used as support for the manuscript's conclusions. This is especially true of SI Figs. 2 and 3. I referred to the figures (and tables) in the SI PDF as much, if not more, than I did to those in the manuscript. Although the figures are overcomplicated, each figure created for this manuscript, including those in the SI, is important for reaching the conclusions of the manuscript. I recognize there may be a limit on the number of figures allowed in an article for this journal. However, the manuscript would be better served if all the figures were included as part of the main manuscript.

3. The environmental data presented on sheet 3 of the SI Excel file was extremely helpful when reading through the manuscript because it answered all the questions about the consistency of the environment that I was asking as I read through the text.

4. Lines 168-173: If the biogeochemical/physical factors are relatively consistent with each other, what heterogeneity are you referring to in the benthic environments that would cause the "significant variations" observed in the composition and distribution of brGDGTs in the sediment? Adding discussion on this may be helpful for your argument by providing a guide for your audience to follow as you walk them through your conclusions.

5. Results and Discussion Section - this section of the manuscript blends the results and discussion of your interpretations so well that I often reread paragraphs to find the raw data you were presenting. I suggest adding a data table to the manuscript summarizing the data you want the audience to track as you walk through the discussion, even if it is the mean and standard deviation in the chart (i.e., lines 111-119). Doing so will allow the audience to reference your data easily while reading through the discussion.

Reviewer #2 (Remarks to the Author):

In this manuscript Xiao et al. present a dataset of sediments from three deep-sea regions where brGDGTs were measured. Through a series of correlation analyses between environmental parameters measured on site, or from previously published studies, the authors show that the diffusive oxygen uptake (DOU) parameter best correlates with changes in the brGDGT concentrations, and suggest a correlation between this parameter and the activity from the brGDGT producing organisms. The authors propose the Isomerization and Methylation index of Branched Tetraethers (IMBT) as a metric to estimate

DOU in sediments. This correlation between DOU and IMBT is then shown to be maintained when the study is expanded to previously published and globally distributed marine sediments.

The authors present an interesting and compelling argument about the influence of DOU on the production of brGDGTs in deep-sea environments, where parameters usually associated with GDGTs remain stable (i.e. temperature). I think this is a well written and compelling manuscript that is suited for this journal. I think that only some relatively minor issues need to be tackled first.

My main concern with the manuscript is regarding the introduction of the IMBT index, which the authors show to have a correlation with DOU. While the correlation between brGDGTs and DOU is consistent with the data, the formulation of IMBT strongly resembles that of the Isomer Ratio (IR). Since $IR = (IIa' + IIb' + IIc' + IIIa' + IIIb' + IIIc') / (Ia + IIa' + IIb + IIb' + IIc + IIc' + IIIa + IIIa' + IIIb + IIIb' + IIIc + IIIc')$ and the presented environments already have low abundances of the brGDGT with cyclopentane rings, the equation ends up resembling a lot the proposed IMBT index $IMBT = -\log_{10}((IIIa' + IIa') / (IIIa + IIIa' + IIa + IIa' + I))$ which mostly adds Ia to the denominator, particularly since the authors show that the non-logarithmic formulation of IMBT shows comparable results. Although I think it is possible that the IMBT index does provide new information compared with IR, or any other index, I think that a proper comparison between the performance of IMBT with previously established indices would be important, particularly when the indices are formulated based on correlations and not mechanistic principles.

Detailed comments

Line 53: Feels like a citation is missing here from the sentence.

Line 148 (and others): Why is the index presented as $\sum IIIa / \sum IIa$? Xiao et al., 2016 presents the index as III / IIa .

Reviewer #3 (Remarks to the Author):

Dear editor,

I have completed my review of the manuscript "Strong linkage between benthic oxygen uptake and bacterial tetraether lipids in deep-sea trench regions". It is an interesting manuscript, aiming to explain variation in brGDGTs in marine sediments with contrasting sediment oxygen properties. The manuscript encompasses an interesting dataset, is well written, well-referenced and presents clear figure. Furthermore, the development of a proxy that is mechanistically linked to DOU, would be very valuable for the paleoclimate community. While I support the empirical link between DOU and brGDGT fractional abundances, the interpretation of this mechanism is currently not strong, which I attribute to the lack of concentration data presented in the manuscript. I have outlined several comments below that can improve the manuscript.

Major comments:

While the distribution (i.e. fractional abundance changes) is linked to DOU, the proposed mechanism that causes this is poorly constrained. This is caused in part, by the lack of use of concentration data. The complete discussion is currently built up on correlations between fractional abundance and DOU, which means that we don't know whether we are observing an increase in 5-methyl brGDGTs, or instead a decrease in 6-methyl brGDGTs with increasing DOU values. Determining this would be a necessary step towards making the discussion on mechanisms more clear. Importantly, the method sections describes that quantification was done, but no concentrations are reported. Using concentration data will also allow to evaluate whether DOU is really the main driving factor, or whether TOC/TN, NPP and bottom water oxygen concentration perhaps play a more significant role explaining brGDGT concentrations.

A second major comment is, that if a formula is proposed to link brGDGT distribution to DOU, this formula should include an estimate of error. Also, is the log transformation necessary? The performance of this novel ratio in comparison with existing ratios (DC', IR) should be compared. A figure that plots the most common ratios (MBT'5ME, DC', IR) against DOU would be helpful for the community to interpret manuscripts that report these data.

From time to time, the authors make a statement that is not backed up by data or references. For instance at L 53 ("recent advancements ... marine sediments), L 168 (...exhibited minimal differences), L 296 (... casts no doubt on this notion))

Minor comments:

L 55. Isomerization degree is not a valid term (in the end all 15 brGDGTs are 'isomers'), this needs to be explained as a change from 5-methyl to 6-methyl forms (or vice versa) of penta- and hexamethylated brGDGTs.

L 124. 0-ring brGDGT compound is not a correct term.

L 136. As far as I saw the definition of hadal and non-hadal sites at your setting was not explained yet. Please introduce these terms before.

L 163-165, is repetition, can be removed or tied in with the end of the paragraph (as part of the conclusion).

At L 182-185 the discussion on CBT5ME, reconstructed pH and bottom water pH is very unclear. The point that the acidic bias between reconstructed and measured pH is supposed to make is not clear to me?

L 185. Specify that this is fractional abundance of brGDGTs.

L 192. Before reaching this conclusion, changes in concentration of brGDGTs with DOU and other parameters should also be evaluated.

L 312. Here the authors discuss the mechanism as being related to increased methylation, but what is the correlation between MBT'5ME (degree of methylations) and DOU?

L 325. If nutrient limitations are the proposed mechanism, the link between nutrient availability and

DOU needs to be described better.

L 453. The $\Sigma IIIa/\Sigma IIa$ ratio sometimes also includes the late eluting isomers of IIIa and IIa (so-called IIIa''' and IIa''' or IIIa7ME and IIa7ME). Can the authors specify that their calculation, and that of the compiled data of course, both didn't include these late eluting isomers?

Responses to Reviewers

Reviewer #1 (Remarks to the Author):

Comments:

The manuscript titled “Strong linkage between benthic oxygen uptake and bacterial tetraether lipids in deep-sea trench regions” presents data on the distribution of branched glycerol dialkyl glycerol tetraethers (brGDGTs) in sediment cores from the Atacama, Kermadec, and Mariana trench regions. Within these regions, terrestrial input is limited, pressure is high, and bottom waters are consistently similar temperatures, which allows the authors to make general arguments in the manuscript regarding the relationship between oxygen and brGDGTs in marine sediments. The authors identify a strong relationship between diffusive oxygen uptake, which measures the microbial diagenetic intensity.

The manuscript is well written, and the sample and data collection methods follow the typical methodologies used to obtain this type of data. The data presented is novel, and the implications are intriguing as the data may explain why previous studies of brGDGTs distribution in sedimentary environments have yielded conflicting results and inferences about the favorable conditions for bacterial synthesis of brGDGTs and the role oxygen availability plays on the production and distribution of these biomarkers. The paper should be published; however, I suggest a few improvements to the manuscript below.

Response: We greatly appreciate your constructive feedback and the positive evaluation, which have significantly enhanced the clarity and readability of our manuscript. In response to your suggestions, we have 1) revised Fig. 2 to make it more intuitive and easier to interpret; 2) transferred the correlation analysis plot from the supplementary materials to the main text to provide readers with more information; 3) added a new table to the supplementary materials, summarizing the key data to facilitate an easier navigation through our discussion; and 4) addressed all other feedback provided. We believe these revisions thoroughly address your concerns and fulfill the publication criteria.

Suggested Edits:

1. Several figures are overcomplicated, making them difficult to read. I also struggled to see how the data presented applied to what was discussed in the text. For example, Fig. 2A is visually appealing, but it was difficult to read and make interpretations from the heat maps. As Fig. 2A is organized, the instinct is to read the heat maps as though they represent increasing DOU across some horizontal continuum, like along a cross-section. Instead, the two rows are unrelated to each other (the bottom row shows data from the AT collected at different locations in that trench region, whereas, the top row displays the data from one sample from the MT and 4 samples from the KT that were collected from different locations in that trench region). The data are unrelated and not meant to be correlated, nor do they

represent change across one cross-section.

It was also difficult to see that the sediment depth arrows represent depth into the core. After looking at the figure for a while, I rotated the paper so that the brGDGT labels were at the top of the maps and the arrow indicating the sediment thickness pointed down as it would when reading data from cores. It might be easier for an audience to read if the heat maps are rearranged to read from top to bottom as you would with data in a core sample. There should also be a way to indicate that these maps are not meant to correlate with each other (as you would when creating a fence diagram or correlating strata).

Response: Thank you for your insightful suggestion. We have made two revisions to improve the clarity of Fig. 2. 1) Initially, the horizontal arrangement of different images was based on the DOU levels, but this arrangement corresponded to three different trenches separately. In the revised figure, we have reorganized the figure to present the sampling sites from the three trenches in a unified manner that aligns with DOU gradients, to more clearly show how brGDGT compositions vary with DOU. 2) The original figure's horizontal axis represented sediment depth, which contradicts the common intuition that the vertical axis should represent sediment depth. Therefore, we have rotated the images to make the vertical axis represent sediment depth. Additionally, we have refined the legend to more clearly state that different color blocks visualize the fractional abundance of brGDGTs.

Revision:

Revised Figure 2A: Heat maps showing the fractional abundance of brGDGT compounds in sediment cores from the Mariana, Kermadec and Atacama trench regions. The colormap visualizes the fractional abundance of brGDGTs in each sample. DOU values are labeled for each station site.

Unfortunately, I am still unable to identify the relationship you are describing when you reference the figure in line 143. There may not be a better way to convey this information in a

figure, which can be challenging. I frequently referred to the data chart in the SI material to gain an understanding of what was being displayed in the figure.

Response: Thank you for highlighting this issue. Indeed, Fig. 2 may not have effectively captured the changes in methylation and isomerization degrees of brGDGTs, since it primarily showcased variations in the fractional abundance of different brGDGT compounds. Recognizing this, we have revised the text from "brGDGT methylation and isomerization degrees" to "brGDGT compounds, especially regarding the number of methyl groups and isomeric forms..." Figures 2A and 3A detail the changes in the fractional abundance of brGDGTs and their relationship with DOU. We believe that by combining these two figures, we can more clearly convey the interrelationship between DOU and brGDGT compounds.

Revision:

Lines 145 – 148: The variations in major brGDGT compounds, especially regarding the number of methyl groups and isomeric forms, are closely associated with DOU variations (Fig. 2A and 3A), hinting at a tight interrelationship.

There is a similar issue with the correlation maps in SI Fig. 3. The figure is responsible for displaying too much information and requires an extended amount of time to decipher what is being displayed. I found the SI table with the environmental data helpful as I looked through this figure.

Response: Thank you for your valuable suggestion. We acknowledge that Fig. S3 presented a dense compilation of information, including a array of correlation analyses across different trenches and depth levels. Our intention was to provide detailed evidence to support the robustness of our findings. In response to your suggestion, we have divided Fig. S3 into two parts: one part has been integrated into the main text to complement our discussion, while the other part remains in the supplementary materials. This adjustment helps to prevent information overload. Furthermore, as you pointed out, the Supplementary Dataset is important for understanding the figures, so we have highlighted references to the Supplementary Dataset at key points in the text.

Revision:

Revised Figure 4. Correlation maps between brGDGTs and environmental variables.

Pearson correlation analysis was utilized to assess the relationships between brGDGTs, both in terms of fractional abundances and concentrations (conc.), and a range of environmental variables. These variables include latitude, longitude, water depth, sediment depth, DOU, dissolved oxygen concentrations in bottom water and sediment, annual sea surface and bottom temperatures, NPP, TOC, TN, and TOC/TN ratio (Supplementary Dataset). Notice a logarithmic scale of DOU. The colormap indicates the correlation coefficients (r values), with red indicating positive correlations and blue indicating negative correlations. Statistical significance is denoted as $*p < 0.05$. Due to the limited number of cores from the KT ($n = 4$) and MT ($n = 1$), conducting separate correlation analyses for these trenches was impractical.

Alternatively, we performed correlation analyses on samples from the AT, combined samples from the AT and KT, and combined samples from all the trenches. While analyses for surface samples are included here, those for the average values of each core and for the entire core sample dataset are presented in Fig. S3.

Revised Figure S3:

2. The figures currently in the supplemental information should be included in the main manuscript because they were crucial to understanding the other images in your paper, and they are cited as evidence used as support for the manuscript's conclusions. This is especially true of SI Figs. 2 and 3. I referred to the figures (and tables) in the SI PDF as much, if not more, than I did to those in the manuscript. Although the figures are overcomplicated, each figure created for this manuscript, including those in the SI, is important for reaching the conclusions of the manuscript. I recognize there may be a limit on the number of figures allowed in an article for this journal. However, the manuscript would be better served if all the figures were included as part of the main manuscript.

Response: Thank you for pointing this out. The figures in the supplementary materials are indeed valuable in supporting the manuscript's conclusions. Following your suggestion and

considering the journal's general format, we have strategically relocated a portion of Fig. S3 to the main text. This aims to make critical data, which directly supports our findings, more readily accessible to readers within the main narrative. As a result, the total number of figures in the main text has reached six, which we believe is a suitable number that conveys necessary information while avoiding an excess of figures.

Revision: Please refer to the revised Fig. 4 and S3 to see the changes.

3. The environmental data presented on sheet 3 of the SI Excel file was extremely helpful when reading through the manuscript because it answered all the questions about the consistency of the environment that I was asking as I read through the text.

Response: We appreciate your positive feedback. We have added references to the Supplementary Dataset at strategic points within the main text, aiming to guide readers towards these additional resources and facilitating a better engagement with the manuscript's content.

Revision:

Lines 113 – 114: All supporting data are detailed in the Supplementary Dataset, with the principal data summarized in Table S1.

4. Lines 168-173: If the biogeochemical/physical factors are relatively consistent with each other, what heterogeneity are you referring to in the benthic environments that would cause the “significant variations” observed in the composition and distribution of brGDGTs in the sediment? Adding discussion on this may be helpful for your argument by providing a guide for your audience to follow as you walk them through your conclusions.

Response: Thank you for bringing this issue to our attention. In the revised manuscript, we have omitted the term "significant heterogeneity" to avoid any confusion. Also, we have specified that it is the DOU that we pinpoint as a critical driver of the observed significant variations in the composition and distribution of brGDGTs. This specificity helps to clarify our argument and guide readers a clearer pathway through our discussion.

Revision:

Lines 172 – 175: Such findings imply that brGDGTs in the trench sites are likely primarily produced within the sediments; and the variability in brGDGTs likely reflect the conditions of benthic environments, such as benthic DOU, a topic that will be discussed in the following section.

5. Results and Discussion Section - this section of the manuscript blends the results and discussion of your interpretations so well that I often reread paragraphs to find the raw data

you were presenting. I suggest adding a data table to the manuscript summarizing the data you want the audience to track as you walk through the discussion, even if it is the mean and standard deviation in the chart (i.e., lines 111-119). Doing so will allow the audience to reference your data easily while reading through the discussion.

Response: We have carefully considered your valuable suggestion. In the revised manuscript, we have added a table into the supplementary materials, summarizing the main data results. This table is designed to simplify the reader's navigation through our findings. Additionally, we have referenced this table and the Supplementary Dataset at strategic points in the "Results and Discussion" section.

Revision:

Lines 113 – 114: All supporting data are detailed in the Supplementary Dataset, with the principal data summarized in Table S1.

Table S1. Summary of main data presented in the "Results and discussion" section of the main text. Main data including sedimentation rates, DOU values, concentrations and fractional abundances of brGDGTs, along with related brGDGT proxies, from sites in the Mariana Trench (MT), Kermadec Trench (KT) and Atacama Trench (AT) regions.

Sedimentation rate data are cited from Glud et al. (2013)⁵, Oguri et al. (2022)⁶, and Zabel et al. (2022)⁷. DOU data are cited from Glud et al. (2013)⁵ and Glud et al. (2021)⁸. Data on brGDGT concentrations, fractional abundances, and related proxies were generated in this study, which have been detailed in the Supplementary Dataset.

Parameter	MT site (n = 1)	KT sites (n = 4)	AT sites (n = 8)
Sedimentation rate (cm yr ⁻¹)	0.04	0.03 – 0.04	0.03 – 0.08
DOU (μmol m ⁻² d ⁻¹)	154	152 – 538	355 – 1793
BrGDGT concentration (ng g ⁻¹)	15±3	12±7	215±170
BrGDGT concentration normalized by TOC (μg g ⁻¹ TOC)	5±1	4±1	36±29
BrGDGT-IIIa (%)	0±0	5±3	10±3
BrGDGT-IIIa' (%)	73±2	32±17	15±9
BrGDGT-IIa (%)	0±0	8±4	14±8
BrGDGT-IIa' (%)	10±1	11±2	8±1
BrGDGT-IIb (%)	0±0	3±2	2±1
BrGDGT-IIb' (%)	1±0	6±2	9±5
BrGDGT-IIc (%)	0±0	1±0	1±0
BrGDGT-IIc' (%)	0±0	2±1	3±2
BrGDGT-Ia (%)	12±1	21±5	29±4
BrGDGT-Ib (%)	2±0	6±3	5±1
BrGDGT-Ic (%)	1±0	3±1	3±1
Non-cyclized brGDGTs (%)	96±1	78±8	76±7
Mono-cyclized brGDGTs (%)	2±1	16±6	17±5
Di-cyclized brGDGTs (%)	1±0	6±2	7±2
6-methyl brGDGTs (%)	100±0	73±15	56±19
5-methyl brGDGTs (%)	0±0	27±15	44±19

Hexamethylated brGDGTs (%)	73±2	37±14	26±7
Pentamethylated brGDGTs (%)	11±1	32±6	38±4
Tetramethylated brGDGTs (%)	15±2	31±8	37±4
BIT	0.03±0.01	0.07±0.05	0.07±0.05
ΣIIIa/ΣIIa	7.14±0.98	2.00±1.03	1.45±0.91
IR	1.00±0.00	0.73±0.15	0.56±0.19
CBT'	0.84±0.04	0.21±0.29	-0.16±0.29
MBT' _{5ME}	1.00±0.00	0.67±0.10	0.59±0.10
CBT _{5ME}	0.81±0.10	0.55±0.15	0.73±0.15
DC'	0.10±0.02	0.27±0.06	0.26±0.10
IMBT	0.06±0.01	0.30±0.14	0.55±0.19

Reviewer #2 (Remarks to the Author):

In this manuscript Xiao et al. present a dataset of sediments from three deep-sea regions where brGDGTs were measured. Through a series of correlation analyses between environmental parameters measured on site, or from previously published studies, the authors show that the diffusive oxygen uptake (DOU) parameter best correlates with changes in the brGDGT concentrations, and suggest a correlation between this parameter and the activity from the brGDGT producing organisms. The authors propose the Isomerization and Methylation index of Branched Tetraethers (IMBT) as a metric to estimate DOU in sediments. This correlation between DOU and IMBT is then shown to be maintained when the study is expanded to previously published and globally distributed marine sediments.

The authors present an interesting and compelling argument about the influence of DOU on the production of brGDGTs in deep-sea environments, where parameters usually associated with GDGTs remain stable (i.e. temperature). I think this is a well written and compelling manuscript that is suited for this journal. I think that only some relatively minor issues need to be tackled first.

Response: We sincerely appreciate your encouraging comments and constructive feedback on our manuscript. Your suggestion to compare IMBT with other major indexes is incredibly constructive. It has not only deepened our discussion but also offered a richer context for other studies utilizing these classic indexes. We have made revisions based on all your suggestions. We trust that these revisions have improved the manuscript's quality and align with the publication criteria.

My main concern with the manuscript is regarding the introduction of the IMBT index, which the authors show to have a correlation with DOU. While the correlation between brGDGTs and DOU is consistent with the data, the formulation of IMBT strongly resembles that of the Isomer Ratio (IR). Since $IR = \frac{(IIa' + IIb' + IIc' + IIIa' + IIIb' + IIIc')}{(IIa + IIa' + IIb + IIb' + IIc + IIc' + IIIa + IIIa' + IIIb + IIIb' + IIIc + IIIc')}$ and the presented environments already have low abundances of the brGDGT with cyclopentane rings, the equation ends up resembling a lot the proposed IMBT index $IMBT = -\log_{10} \left(\frac{IIIa' + IIa'}{IIIa + IIIa' + IIa + IIa' + I} \right)$ which mostly adds Ia to the denominator, particularly since the authors show that the non-logarithmic formulation of IMBT shows comparable results. Although I think it is possible that the IMBT index does provide new information compared with IR, or any other index, I think that a proper comparison between the performance of IMBT with previously established indices would be important, particularly when the indices are formulated based on correlations and not mechanistic principles.

Response: We appreciate your insightful suggestion regarding the comparison between IMBT and other indexes. In the revised manuscript, we have conducted a comprehensive comparison between IMBT with other widely used brGDGT indexes, including MBT'_{5ME}, CBT'_{5ME}, DC' (degree of cyclization), IR, and CBT'. Since IMBT reflects both methylation and isomerization degrees of brGDGTs, it naturally shares similarities and exhibits strong

correlations with these traditional isomerization and methylation indexes (e.g., MBT'_{5ME} and IR). However, it is important to highlight that IMBT uniquely captures both methylation and isomerization aspects and shows the strongest correlation with DOU. We have also compared IMBT with cyclization indexes (e.g., CBT_{5ME} and DC'). These discussions aim to clarify IMBT's utility in reflecting DOU and assist the academic community in better interpreting our results.

Revision:

Revised Figure S5. Correlation maps between brGDGT indexes and DOU. Pearson correlation analysis was employed to assess the relationships between multiple brGDGT indexes and DOU. The examined indexes include IMBT, MBT'_{5ME}, CBT_{5ME}, DC', IR, and CBT'. The calculations for DC' and CBT' were based on equations from De Jonge et al. (2021)³ and De Jonge et al. (2014)⁴, respectively. Descriptions of the other indexes are provided in the "Methods" section of the main text. Notice a logarithmic scale of DOU. The colormap indicates the correlation coefficients (r values), with red indicating positive correlations and blue indicating negative correlations. Correlation analyses were performed individually for surface sediments, the average values of each core, and the entire core sample dataset. Notably, IMBT, MBT'_{5ME}, IR, and CBT' show strong correlations with DOU, while CBT_{5ME} and DC' exhibit much weaker correlations.

Lines 280 – 291: Furthermore, we compared the IMBT index with other commonly used brGDGT indexes, including MBT'_{5ME}, CBT_{5ME}, DC' (degree of cyclization)⁵⁰, IR, and CBT'¹³ (Fig. S5). The analysis revealed that IMBT strongly correlates with indexes related to brGDGT methylation and isomerization (e.g., MBT'_{5ME} and IR; absolute r values, $|r| > 0.85$), while showing much weaker correlations with cyclization indexes (e.g., CBT_{5ME} and DC'; $|r| < 0.35$). This pattern extends to DOU, which exhibits strong correlations with methylation

and isomerization indexes ($|r| > 0.70$), but only minimal correlations with cyclization indexes ($|r| < 0.30$). Notably, the correlation between DOU and IMBT stands out as the most pronounced among these indexes. Considering the established association of methylation indexes with temperature and isomerization indexes with pH^{13,14}, the influence of DOU is crucial in evaluating the reliability of methylation-based temperature and isomerization-based pH indexes, especially in marine settings characterized by predominant in-situ brGDGT production.

Lines 366 – 370: Our findings underscore the pivotal role of DOU in modulating brGDGT methylation and isomerization patterns, which are typically associated with temperature and pH^{13,14}, respectively. In marine settings where in-situ brGDGT production is predominant, accounting for the effect of DOU becomes essential for evaluating methylation-based temperature and isomerization-based pH indexes^{13,61}.

Detailed comments

Line 53: Feels like a citation is missing here from the sentence.

Response: We have included citations to studies by Xiao et al. (2022), Wu et al. (2021), and Weber et al. (2018) in the revised manuscript.

Revision:

Lines 52 – 54: Recent advancements in brGDGT research have unveiled potential connections between these biomarkers and benthic oxygen conditions in sediments¹⁰⁻¹².

Line 148 (and others): Why is the index presented as $\Sigma\text{IIIa}/\Sigma\text{IIa}$? Xiao et al., 2016 presents the index as III/IIa .

Response: Thank you for pointing out this confusion. The original IIIa/IIa ratio by Xiao et al. (2016) indeed included both 5- and 6-methyl brGDGTs. However, at the time of their study, the distinction between 5- and 6-methyl brGDGTs was not commonly made in the literature. Subsequent research adopted the notation $\Sigma\text{IIIa}/\Sigma\text{IIa}$ to differentiate between 5- and 6-methyl brGDGTs more clearly. We have provided an explanation of this in the revised manuscript.

Revision:

Lines 476 – 479: The $\Sigma\text{IIIa}/\Sigma\text{IIa}$ index was calculated following the definition of the IIIa/IIa ratio by Xiao et al. (2016)¹⁸, which combined the 5- and 6-methyl isomers to correspond to traditional chromatographic approaches^{13,14}.

References: All references cited can be found in the main text.

Reviewer #3 (Remarks to the Author):

I have completed my review of the manuscript “Strong linkage between benthic oxygen uptake and bacterial tetraether lipids in deep-sea trench regions”. It is an interesting manuscript, aiming to explain variation in brGDGTs in marine sediments with contrasting sediment oxygen properties. The manuscript encompasses an interesting dataset, is well written, well-referenced and presents clear figure. Furthermore, the development of a proxy that is mechanistically linked to DOU, would be very valuable for the paleoclimate community. While I support the empirical link between DOU and brGDGT fractional abundances, the interpretation of this mechanism is currently not strong, which I attribute to the lack of concentration data presented in the manuscript. I have outlined several comments below that can improve the manuscript.

Response: We sincerely appreciate your insightful review and positive feedback. Your comments have been instrumental in refining our manuscript. In response to your suggestions, we have 1) incorporated detailed discussion on the correlations between brGDGT concentrations and environmental factors (including DOU) and expanded the discussion on the mechanisms of interaction between IMBT and DOU; 2) compared IMBT with other commonly used brGDGT indexes; 3) added references at strategic points to support our arguments; and 4) carefully addressed all the minor comments. We believe that these revisions improve the manuscript's quality and align with the publication criteria.

Major comments:

While the distribution (i.e. fractional abundance changes) is linked to DOU, the proposed mechanism that causes this is poorly constrained. This is caused in part, by the lack of use of concentration data. The complete discussion is currently built up on correlations between fractional abundance and DOU, which means that we don't know whether we are observing an increase in 5-methyl brGDGTs, or instead a decrease in 6-methyl brGDGTs with increasing DOU values. Determining this would be a necessary step towards making the discussion on mechanisms more clear. Importantly, the method sections describes that quantification was done, but no concentrations are reported. Using concentration data will also allow to evaluate whether DOU is really the main driving factor, or whether TOC/TN, NPP and bottom water oxygen concentration perhaps play a more significant role explaining brGDGT concentrations.

Response: Thank you for your constructive suggestion. We have actually measured the brGDGT concentrations, briefly mentioned the concentration data early in the "Results and Discussion" section and included the corresponding data in the Supplementary Dataset. However, our initial discussion focused on exploring the relationship between the fractional abundance of brGDGTs and environmental parameters, without discussing the concentration of brGDGTs. In the revised manuscript, we have included a comprehensive correlation analysis that includes brGDGT concentrations. This analysis revealed relationships between DOU, NPP, TOC, and brGDGT concentrations. We further explained these relationships in

our discussion on mechanisms: lower NPP and TOC may indicate a scarcity of organic carbon supply, which corresponds to lower brGDGT concentrations; simultaneously, a reduction in organic carbon supply could be associated with decreased microbial activity, thereby leading to lower DOU levels. This links lower DOU with lower brGDGT concentrations. These new findings enhance the discussion and bolster the manuscript's arguments. We hope that this more analysis addresses your suggestions.

Revision:

Lines 207 – 208: Our correlation analysis reveals relationships between DOU, NPP, TOC and brGDGT concentrations.

Lines 339 – 343: In the deep ocean, DOU levels typically reflect microbial diagenetic intensity^{3,39}. The correlations observed across our study sites between NPP, TOC, DOU and brGDGT concentrations (Fig. 4 and S3) suggest a likely connection between a decreased supply of organic carbon and lower concentration of brGDGTs, alongside reduced DOU levels—indicative of diminished microbial diagenetic activity.

Revised Figure 4. Correlation maps between brGDGTs and environmental variables.

Revised Figure S3:

A second major comment is, that if a formula is proposed to link brGDGT distribution to DOU, this formula should include an estimate of error. Also, is the log transformation necessary? The performance of this novel ratio in comparison with existing ratios (DC', IR) should be compared. A figure that plots the most common ratios (MBT'_{5ME}, DC', IR) against DOU would be helpful for the community to interpret manuscripts that report these data.

Response: Thank you for this valuable comment. In the revised manuscript, we have made the following improvements: 1) We have included the estimated error associated with the IMBT calculation; 2) We have expanded the discussion comparing IMBT with other commonly used indexes, including MBT'_{5ME}, CBT'_{5ME}, DC' (degree of cyclization), IR, and CBT'. Since IMBT reflects both methylation and isomerization degrees of brGDGTs, it naturally exhibits strong correlations with these traditional isomerization and methylation indexes (e.g., MBT'_{5ME} and IR). Accordingly, DOU also shows strong correlations with these indexes. We also compared IMBT with cyclization indexes (e.g., CBT'_{5ME} and DC'). These discussions will assist readers in better interpreting our results. 3) We have explained the rationale behind opting for logarithmic transformation. While our data showed that both logarithmic and linear analyses yielded highly similar results, the choice to use logarithmic representation was deemed more appropriate across a wider range of DOU values.

Revision:

Lines 249 – 250: The IMBT index shows a significant positive correlation with DOU ($r = 0.88$, $RMSE = 400 \mu\text{mol m}^{-2} \text{d}^{-1}$, $p < 0.001$, Fig. 5A, equation 2).

Lines 280 – 291: Furthermore, we compared the IMBT index with other commonly used brGDGT indexes, including MBT'_{5ME}, CBT'_{5ME}, DC' (degree of cyclization)⁵⁰, IR, and CBT'¹³ (Fig. S5). The analysis revealed that IMBT strongly correlates with indexes related to brGDGT methylation and isomerization (e.g., MBT'_{5ME} and IR; absolute r values, $|r| > 0.85$), while showing much weaker correlations with cyclization indexes (e.g., CBT'_{5ME} and DC'; $|r| < 0.35$). This pattern extends to DOU, which exhibits strong correlations with methylation and isomerization indexes ($|r| > 0.70$), but only minimal correlations with cyclization indexes ($|r| < 0.30$). Notably, the correlation between DOU and IMBT stands out as the most pronounced among these indexes. Considering the established association of methylation indexes with temperature and isomerization indexes with pH^{13,14}, the influence of DOU is crucial in evaluating the reliability of methylation-based temperature and isomerization-based pH indexes, especially in marine settings characterized by predominant in-situ brGDGT production.

Lines 366 – 370: Our findings underscore the pivotal role of DOU in modulating brGDGT methylation and isomerization patterns, which are typically associated with temperature and pH^{13,14}, respectively. In marine settings where in-situ brGDGT production is predominant, accounting for the effect of DOU becomes essential for evaluating methylation-based temperature and isomerization-based pH indexes^{13,61}.

Revised Figure S5. Correlation maps between brGDGT indexes and DOU. Pearson correlation analysis was employed to assess the relationships between multiple brGDGT indexes and DOU. The examined indexes include IMBT, MBT'_{5ME}, CBT'_{5ME}, DC', IR, and CBT'. The calculations for DC' and CBT' were based on equations from De Jonge et al. (2021)³ and De Jonge et al. (2014)⁴, respectively. Descriptions of the other indexes are provided in the "Methods" section of the main text. Notice a logarithmic scale of DOU. The colormap indicates the correlation coefficients (r values), with red indicating positive correlations and blue indicating negative correlations. Correlation analyses were performed individually for surface samples, the average values of each core, and the entire core sample dataset. Notably, IMBT, MBT'_{5ME}, IR, and CBT' show strong correlations with DOU, while CBT'_{5ME} and DC' exhibit much weaker correlations.

From time to time, the authors make a statement that is not backed up by data or references. For instance at L 53 (“recent advancements ... marine sediments), L 168 (...exhibited minimal differences), L 296 (... casts no doubt on this notion)).

Response: Thank you for pointing out these matters. We have carefully reviewed our manuscript and revised it by incorporating relevant references at the corresponding points to support our arguments. Additionally, we have included additional references to our Supplementary Dataset.

Revision:

Lines 52 – 54: Recent advancements in brGDGT research have unveiled potential connections between these biomarkers and benthic oxygen conditions in sediments¹⁰⁻¹².

Lines 169 – 171: ...the biogeochemical/physical factors like temperature, salinity, dissolved

oxygen concentration, and pH in the water column exhibited minimal differences^{44,45}.

Lines 313 – 316: However, the elevated IMBT values in trench sediments, which experience significantly higher hydrostatic pressures compared to nearby abyssal sediments (e.g., KT6 vs. KT7, and AT3 vs. AT7; Supplementary Dataset), cast doubt on this notion.

Minor comments:

L 55. Isomerization degree is not a valid term (in the end all 15 brGDGTs are 'isomers'), this needs to be explained as a change from 5-methyl to 6-methyl forms (or vice versa) of penta- and hexamethylated brGDGTs.

Response: We have adjusted the phrasing accordingly.

Revision:

Lines 55 – 57: They exhibit 15 common unique structures, characterized by variances in methylation and cyclization degrees, as well as by their presence in different isomeric forms (e.g., 5- and 6-methyl brGDGTs; Fig. S1)¹³⁻¹⁵.

L 124. 0-ring brGDGT compound is not a correct term.

Response: We have corrected the term "0-ring brGDGTs" to "non-cyclized brGDGTs" and adjusted the relevant descriptions.

Revision:

Lines 127 – 130: In all three trench regions, the non-cyclized brGDGT compounds (IIIa, IIIa', IIa, IIa', and Ia) were predominant. They constituted 96±1% in the MT, 78±8% in the KT, and 76±7% in the AT (Fig. 2A; Supplementary Dataset). In contrast, the mono-cyclized (IIb, IIb', and Ib) and di-cyclized brGDGTs (IIc, IIc', and Ic)...

L 136. As far as I saw the definition of hadal and non-hadal sites at your setting was not explained yet. Please introduce these terms before.

Response: In the revised manuscript, we have clarified that "non-hadal sites" refer to sites with water depths less than 6000 m, while "hadal sites" refer to sites with water depth exceeds 6000 m. We have also provided the appropriate references for support. Furthermore, the Supplementary Dataset have indicated which sites are hadal sites and which are non-hadal sites.

Revision:

Lines 139 – 140: At the intra-trench scale, notable differences were observed between non-hadal sites (depth < 6000 m) and hadal sites (depth > 6000 m⁴²).

L 163-165, is repetition, can be removed or tied in with the end of the paragraph (as part of the conclusion).

Response: We appreciate your guidance on this matter. We have removed this sentence to the end of the paragraph and closely integrated it with our arguments.

Revision:

Lines 175 – 178: ...This inference aligns with a growing body of research suggesting that marine-derived brGDGTs present in sediments predominantly originates from benthic bacteria inhabiting the sediments, particularly in deep-sea settings^{10,23}.

At L 182-185 the discussion on CBT_{5ME}, reconstructed pH and bottom water pH is very unclear. The point that the acidic bias between reconstructed and measured pH is supposed to make is not clear to me?

Response: Thank you for the comment. We have revised the relevant sentences and adjusted the way the discussion is presented.

Revision:

Lines 186 – 190: Despite the absence of in-situ sediment porewater pH measurements, pH estimates derived from the CBT_{5ME} index (AT, 5.9 – 7.2, 6.6±0.3; KT, 6.2 – 7.2, 6.9±0.3; MT, 6.1 – 6.1, 6.4±0.2) show an acidic deviation from the typically alkaline pH of hadal bottom seawater and sediment porewater^{44,46,47}, suggesting that pH might not be the main influence on brGDGTs.

L 185. Specify that this is fractional abundance of brGDGTs.

Response: We have revised it accordingly.

Revision:

Lines 190 – 192: ...we performed correlation analyses comparing both the fractional abundance and concentration of brGDGTs with multiple parameters.

L 192. Before reaching this conclusion, changes in concentration of brGDGTs with DOU and other parameters should also be evaluated.

Response: Thank you for emphasizing this point. In the revised manuscript, we have expanded our correlation analysis to incorporate brGDGT concentrations. Please refer to our response to major comment [1] for a more detailed explanation.

L 312. Here the authors discuss the mechanism as being related to increased methylation, but what is the correlation between MBT'5ME (degree of methylations) and DOU?

Response: Thank you for emphasizing this point. In the revised manuscript, we have thoroughly compared the relationships between DOU and other commonly used brGDGT indexes. Please refer to our response to major comment [2] for a more detailed explanation.

L 325. If nutrient limitations are the proposed mechanism, the link between nutrient availability and DOU needs to be described better.

Response: Thank you for highlighting this confusion. We have further refined our discussion to elucidate the link between nutritional limitations and DOU more clearly. Please note that we used the term "nutritional" which refers to the bioavailability and utilization of nutrients by organisms, rather than "nutrient".

Revision:

Lines 339 – 343: In the deep ocean, DOU levels typically reflect microbial diagenetic intensity^{3,39}. The correlations observed across our study sites between NPP, TOC, DOU and brGDGT concentrations (Fig. 4 and S3) suggest a likely connection between a decreased supply of organic carbon and lower concentration of brGDGTs, alongside reduced DOU levels—indicative of diminished microbial diagenetic activity.

Lines 347 – 350: Hence, we propose that increased methylation and isomerization of brGDGTs may enhance the fitness of source organisms in deep-sea habitats under nutritional limitations, which usually coincide with reduced microbial activity and lower DOU levels.

L 453. The $\Sigma\text{IIIa}/\Sigma\text{IIa}$ ratio sometimes also includes the late eluting isomers of IIIa and IIa (so-called IIIa''' and IIa''' or IIIa7ME and IIa7ME). Can the authors specify that their calculation, and that of the compiled data of course, both didn't include these late eluting isomers?

Response: Yes, we only included 5- and 6-methyl compounds when calculating the $\Sigma\text{IIIa}/\Sigma\text{IIa}$ ratio and did not include 7-methyl compounds. We have made this clear in the revised manuscript.

Revision:

Lines 476 – 480: The $\Sigma\text{IIIa}/\Sigma\text{IIa}$ index was calculated following the definition of the IIIa/IIa ratio by Xiao et al. (2016)¹⁸, which combined the 5- and 6-methyl isomers to correspond to traditional chromatographic approaches^{13,14}. Note that the calculation of $\Sigma\text{IIIa}/\Sigma\text{IIa}$ values excludes the 7-methyl isomers⁸⁰.

References: All references cited can be found in the main text.

REVIEWER COMMENTS

Reviewer #1 (Remarks to the Author):

I agree with the authors that all of my concerns from the first review have been addressed in their responses and the final manuscript.

Reviewer #2 (Remarks to the Author):

Upon reading the modified version of the manuscript by Xiao et al. I find that all the minor revisions have been successfully addressed by the authors.

I appreciate the efforts of the authors to address my comments regarding the comparison between the previously established proxies and IMBT. However I think that the new Figure S5 highlights, I believe, Reviewer #3 and my comment about comparing the advantages of IMBT over older indices. Figure S5 shows a strong correlation between IMBT and both IR, CBT', like the authors discuss in the paper, and this is reflected in similar absolute relationships between these indices and DOU (presented on log scale). What I think is truly missing is a figure akin to figure 5, but with both IR and CBT', comparing the regression with these indices. If IMBT does improve in a statistically significant way, I think just adding these plots as supplementary figures would be enough and I would be happy to recommend the manuscript for publication. I think even if this analysis shows that IMBT does not improve over IR or CBT', this is by no means of detriment to the manuscript, which shows a strong correlation between brGDGTs and DOU, it may just require reconsidering if the IMBT is a necessary index or if IR or CBT' provide the same information.

I want to stress that I think this is a very good and publication-worthy manuscript, I just think that this relatively simple test will strengthen the author's point and either highlight the importance of IMBT or streamline the use of brGDGTs to investigate DOU in these settings.

Responses to Reviewers

Reviewer #1 (Remarks to the Author):

I agree with the authors that all of my concerns from the first review have been addressed in their responses and the final manuscript.

Response: We appreciate your feedback and recommendation for publication. Thank you for dedicating your time to review our work.

Reviewer #2 (Remarks to the Author):

Upon reading the modified version of the manuscript by Xiao et al. I find that all the minor revisions have been successfully addressed by the authors.

I appreciate the efforts of the authors to address my comments regarding the comparison between the previously established proxies and IMBT. However I think that the new Figure S5 highlights, I believe, Reviewer #3 and my comment about comparing the advantages of IMBT over older indices. Figure S5 shows a strong correlation between IMBT and both IR, CBT', like the authors discuss in the paper, and this is reflected in similar absolute relationships between these indices and DOU (presented on log scale). What I think is truly missing is a figure akin to figure 5, but with both IR and CBT', comparing the regression with these indices. If IMBT does improve in a statistically significant way, I think just adding these plots as supplementary figures would be enough and I would be happy to recommend the manuscript for publication. I think even if this analysis shows that IMBT does not improve over IR or CBT', this is by no means of detriment to the manuscript, which shows a strong correlation between brGDGTs and DOU, it may just require reconsidering if the IMBT is a necessary index or if IR or CBT' provide the same information.

I want to stress that I think this is a very good and publication-worthy manuscript, I just think that this relatively simple test will strengthen the author's point and either highlight the importance of IMBT or streamline the use of brGDGTs to investigate DOU in these settings.

Response: We sincerely appreciate your continued engagement and constructive guidance regarding our manuscript. Your recommendation for publication is greatly valued, and in accordance with your latest suggestions, we have carefully revised our work. Following your insights, we have performed additional analyses to assess the correlation of DOU with commonly used brGDGT-based indexes such as IR, CBT', MBT'_{5ME} and CBT'_{5ME}. Notably, the correlation between DOU and IMBT stands out as the most pronounced among the indexes examined. These additional analyses have been included in a new figure, Fig. S6, as you suggested. We have also enhanced the referencing of Fig. S5 and S6 within the main text. We believe these updates comprehensively address your comments.

Revision:

Newly added Figure S6. Correlation between DOU and commonly used brGDGT-based indexes. Scatterplot comparing the **A** IR, **B** CBT', **C** MBT'_{5ME}, and **D** CBT_{5ME} indexes with DOU. Notice a logarithmic scale of DOU. Linear regression line (black) and 95% confidence intervals (gray band) are shown. Statistical significance is denoted as *** $p < 0.001$ and ** $p < 0.01$. The circle, triangle and pentagon symbols with error bars represent the average values with standard deviation of variables for sediment cores from the Mariana, Kermadec, and Atacama trench regions, respectively. The colormap indicates the DOU values of the hadal samples.

Lines 280 – 281: Furthermore, we compared the IMBT index with other commonly used brGDGT indexes, including MBT'_{5ME}, CBT_{5ME}, DC' (degree of cyclization)⁵⁰, IR, and CBT'¹³ (Fig. S5 and S6).

Lines 285 – 288: This pattern extends to DOU, which exhibits strong correlations with methylation and isomerization indexes ($|r| > 0.70$), but only minimal correlations with cyclization indexes ($|r| < 0.30$) (Fig. S5 and S6). Notably, the correlation between DOU and IMBT stands out as the most pronounced among these indexes.

Lines 368 – 371: In marine settings where in-situ brGDGT production is predominant, accounting for the effect of DOU becomes essential for evaluating methylation-based temperature and isomerization-based pH indexes^{13,61} (Fig. S5 and S6).